# Analysis of multi-condition single-cell data with latent embedding multivariate regression

Constantin Ahlmann-Eltze ®[1,2] ✉ & Wolfgang Huber ®[1] ✉

Identifying gene expression differences in heterogeneous tissues across conditions is a fundamental biological task, enabled by multi-condition single-cell RNA sequencing (RNA-seq). Current data analysis approaches divide the constituent cells into clusters meant to represent cell types, but such discrete categorization tends to be an unsatisfactory model of the underlying biology. Here, we introduce latent embedding multivariate regression (LEMUR), a model that operates without, or before, commitment to discrete categorization. LEMUR (1) integrates data from different conditions, (2) predicts each cell's gene expression changes as a function of the conditions and its position in latent space and (3) for each gene, identifies a compact neighborhood of cells with consistent differential expression. We apply LEMUR to cancer, zebrafish development and spatial gradients in Alzheimer's disease, demonstrating its broad applicability.

Premature discretization of continuous variables leads to artifacts in data analysis and a loss of power; yet, it is the dominant approach to deal with the diversity of cell types and states in multi-condition single-cell data. Lähnemann et al.[1] described overcoming the reliance on clustering or (discrete) cell type assignment before downstream analysis as one of the grand challenges in single-cell data analysis.

Single-cell RNA-seq can be used to study the effect of experimental interventions or observational conditions on a heterogeneous set of cells, for example, from tissue biopsies or organoids. Typically, cells from the same sample share the same condition but come from multiple cell types and states (for example, position in a differentiation or cellular aging path, cell cycle, metabolism). The advantage multi-condition single-cell RNA-seq offers over bulk sequencing is the ability to disentangle expression changes between corresponding cells under different conditions from those between cell types or states.

Here, we present a generative model and inference procedure to address three tasks in multi-condition single-cell data analysis: (1) integrate the data into a common latent space, (2) for each cell, predict the expression it would have in any of the conditions and (3) find interesting and statistically significant patterns of differential expression.

For the first task, many methods exist that all share the more or less explicit aim that the variation remaining in the common latent space represents a cell type or a cell state and no longer the external conditions.

However, only some integration methods address the counterfactual prediction of the second task. Harmony[2], Seurat[3] and MNN[4] have no canonical way to map back from positions in the integrated embedding to the gene expression space. By contrast, scVI[5], scGen[6], CPA[7] and CellOT[8] use autoencoders in which the encoder maps from gene expression space to latent space and the decoder maps back in the reverse direction. Latent embedding multivariate regression (LEMUR), instead of learning an encoder and a decoder separately, fits analytically invertible functions to generate condition-specific gene expression values from the integrated embedding.

For the third task, differential expression analysis across conditions, the state of the art is to take an integrated embedding, assign the cells to clusters and find differentially expressed genes separately for each given cluster using methods known from bulk RNA-seq analysis ('pseudobulking')[9,10]. Here, we turn this process around. We employ the LEMUR counterfactual predictions to compute differential expression statistics for each cell and each gene and then select connected sets of cells with consistent differential expression.

[1]Genome Biology Unit, European Molecular Biology Laboratory (EMBL), Heidelberg, Germany. [2]Faculty of Biosciences, Heidelberg University, Heidelberg, Germany. ✉e-mail: constantin.ahlmann@embl.de; wolfgang.huber@embl.org

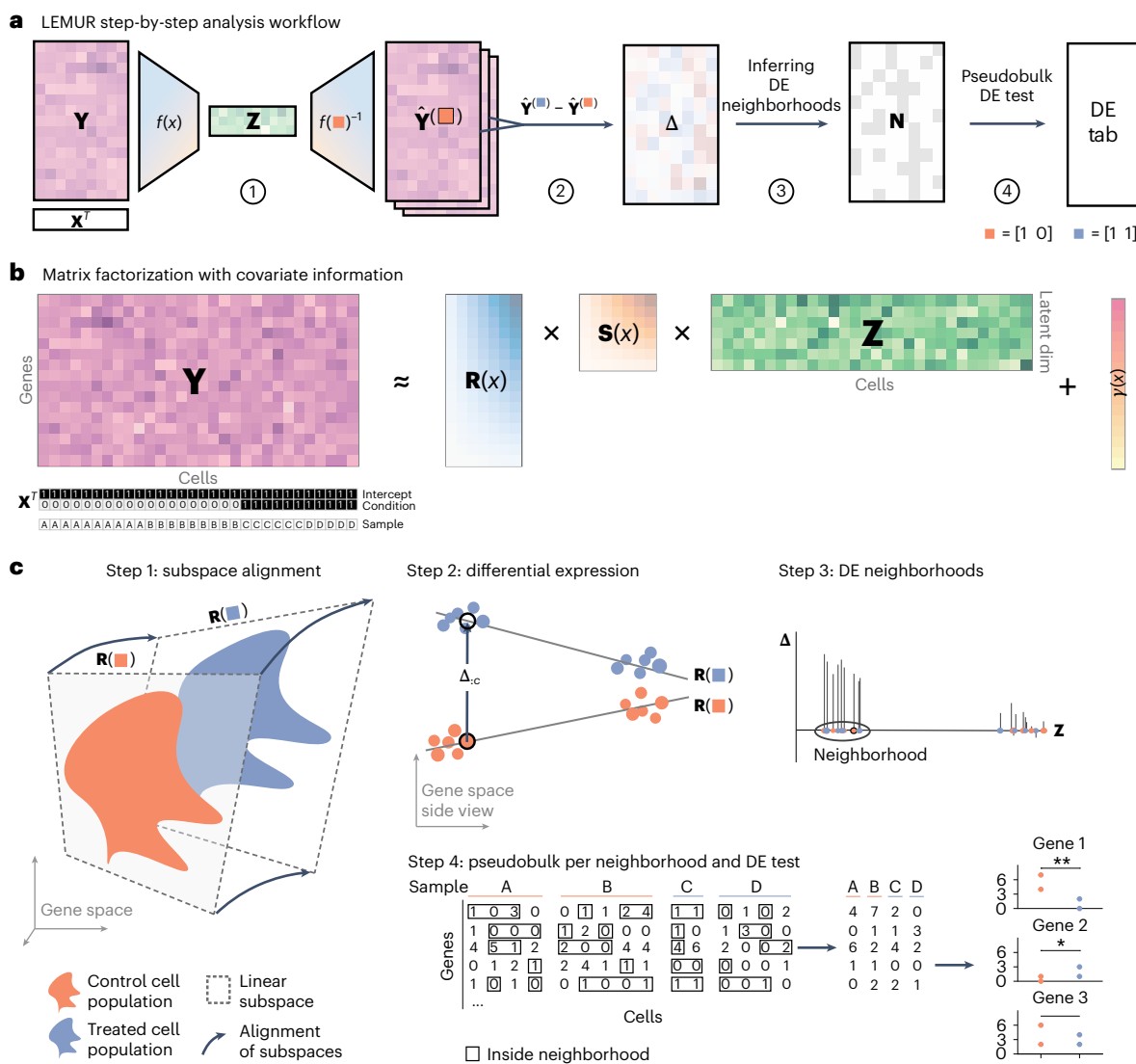

**Fig. 1 | Conceptual overview of LEMUR. a**, Four-step workflow. **b**, The matrix factorization at the core of LEMUR. **c**, Details on each step from **a**: step 1, a linear subspace is fitted separately for each condition. The subspaces for the different conditions are related to each other via affine transformations that are parameterized by the covariates. For this visualization, different two-dimensional subspaces of a three-dimensional gene space are drawn; actual dimensions are higher. Step 2, the differential expression statistic **Δ** is computed as the difference between the predicted values in the control and treated conditions. The visualization shows a top view of the visualization from step 1. Step 3, for each gene, cells close to each other with consistent **Δ** values are grouped into neighborhoods. Step 4, a pseudobulk differential expression test is applied to the cells within each neighborhood. Ctrl., control; DE, differentially expressed; dim, dimension; trt, treatment.

To this end, the LEMUR model decomposes the variation in the data into four sources:

1. The conditions, which are explicitly known,
2. Cell types or states that are not explicitly known but assumed to be representable by a low-dimensional manifold,
3. Interactions between the two and
4. Unexplained residual variability.

LEMUR is implemented in an R package on Bioconductor called lemur and a Python package called pyLemur.

## Results

Fig. 1a outlines the LEMUR workflow. The method takes as input a data matrix of size genes × cells. It assumes that appropriate preprocessing, including size factor normalization and variance stabilizing transformation, was performed[11]. In addition, it expects two tables of metadata for the cells: a categorical variable that, for each cell, specifies the sample (independent experimental unit, for example, tissue biopsy or organoid) it originates from and a design matrix[12].

The design matrix encodes one or more covariates that represent experimental treatments or observational conditions. It is analogous to the design matrix in differential expression tools like limma[13] and DESeq2 (ref. 14) and can account for fully general experimental or study designs. The design matrix can include sources of unwanted variation (for example, experimental batch) and sources of variation whose influence we are interested in (for example, treatment status). We term a unique combination of covariate values a condition. In the simple case of a two-condition comparison, the design matrix is a two-column matrix, the elements of which are all 1 in the first column (intercept) and 0 or 1 in the second column, indicating for each cell which condition it is from.

### Matrix factorization with known covariate information

Consider each cell as a point in a high-dimensional space defined by its measured gene expression profile. The manifold hypothesis posits that

the data concentrate along an (unknown) low-dimensional manifold inside that high-dimensional space. Empirically, it has been found that useful approximations of that low-dimensional manifold can be made by the linear vector space spanned by the first few dozen components of a principal-component analysis (PCA).

The central idea of LEMUR is to represent multi-condition single-cell RNA-seq data using a multi-condition extension of PCA (Fig. 1b). Given a data matrix $\mathbf{Y}$, PCA can be used to approximate $\mathbf{Y} \approx \mathbf{RZ}$ with two smaller matrices: the first, $\mathbf{R}$, contains the top principal components, which act as a basis for the low-dimensional linear subspace; the second, $\mathbf{Z}$, contains the coordinates of each cell with respect to that basis. In this elementary form, there is no place to explicitly encode known experimental or study covariates. LEMUR adds this capability by including a regression analysis component.

Instead of using a single subspace, we find a separate subspace for each condition. For this, we let the subspace-spanning matrix $\mathbf{R}(\mathbf{X})$ depend on the covariates provided in the design matrix $\mathbf{X}$ (Fig. 1c, step 1). With this ansatz, we address the decomposition task posed above: known sources of variation are encoded in $\mathbf{X}$; cell type and state variations are represented by the cell coordinates $\mathbf{Z}$. While $\mathbf{X}$ is explicitly known, $\mathbf{Z}$ is latent; that is, it is estimated from the data. The construction allows modeling interactions, that is, gene expression changes across conditions that are different for different cell types and states. This is the main feature of LEMUR.

A second feature of this construction is between-condition integration: data from cells observed in different conditions are mapped into a common latent space. By default, the integration is based on the alignment of the respective subspaces, but it can optionally be improved by information that indicates that certain cells observed in different conditions correspond to each other and thus should be close to each other in the latent space $\mathbf{Z}$. Such information can come in the form of explicit landmarks, that is, from cells that express distinctive marker genes, or via statistical properties of the cells exploited by methods such as Harmony[2]. We account for such correspondence by adding an affine transformation $\mathbf{S}(\mathbf{X})$ to the model.

A third feature of our model is its ability to predict, for each cell, how its gene expression profile would look like in any of the conditions, even though it was only observed in one of them. In fact, predictions are available not only for those positions in latent space where cells were observed but for all positions, that is, also for hypothetically interpolating or extrapolating cell types and states. We use these predictions to find changes in gene expression that are coordinated across regions of latent space, that is, across the same or similar cell types and states.

Fig. 2 shows a stylized illustration of the LEMUR approach. LEMUR fits one one-dimensional subspace (line) per condition, each parameterized by a rotation applied to a common base space. The parametric model yields a predicted expression value for each cell in each condition, and we look for regions in latent space (here, we have two major regions, left and right) in which predictions are consistently positive or negative. In the Methods, we provide a more formal mathematical specification.

### Cluster-free differential expression analysis

We can predict the expression of a cell in any condition using this parametric model. The differential expression between two conditions (Fig. 1c, step 2) is just the difference between their predictions and can be computed for each cell, even though, for any cell, data were only observed in exactly one replicate of one condition (Fig. 2b).

The resulting matrix of differential expression estimates, $\mathbf{\Delta}$, has two uses: first, we can visualize the differential expression values for each gene as a function of latent space. Typical choices for the dimension of the latent space are ten to 100, and solely for visualization, we use a further nonlinear dimension reduction into two-dimensional scatterplots, such as uniform manifold approximation and projection (UMAP)[15]. Examples are shown in Figs. 4 and 6. Second, we use $\mathbf{\Delta}$

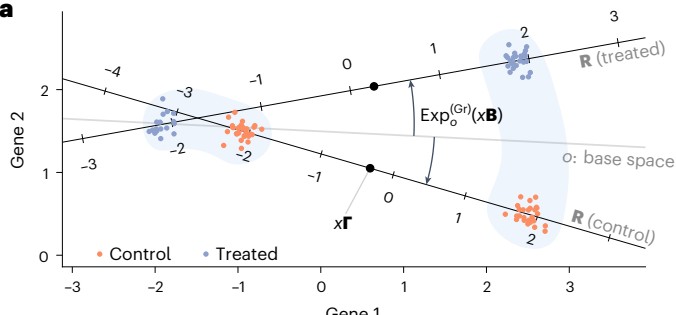

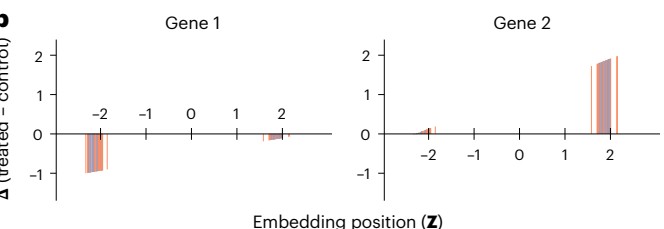

**Fig. 2 | Stylized example with two genes observed in two groups of cells in two conditions. a**, Scatterplot of the gene space with condition-specific one-dimensional subspaces. See the Methods for the mathematical details. **b**, Predicted differential expression of the two genes in each cell. Gene 1 is less expressed in treatment compared to control only in the 'left' cell type; gene 2 is upregulated in the treatment only in the 'right' cell type.

to guide the identification of differential expression neighborhoods, that is, groups of cells that consistently show differential expression for a particular gene (Fig. 1c, step 3). The intention for these neighborhoods is to be connected, convex and maximal, that is, the differential expression pattern would become disproportionately less consistent and less significant if the neighborhood were extended.

To rank and assess our level of confidence in the found neighborhoods, we do not attempt to measure the statistical uncertainty of the predictions $\mathbf{\Delta}$. Instead, we use pseudobulk aggregation[9] of the original data: raw counts, if available, otherwise, the log-normalized values. For each sample, we sum up the original counts or take the mean of the log-normalized values of the cells in the neighborhood to obtain a neighborhood-specific gene × sample table (Fig. 1c, step 4), followed by a differential expression test with glmGamPoi[16], edgeR[17] or limma[13].

### Outputs

LEMUR produces the following outputs:

- A common low-dimensional latent space representation of all cells ($\mathbf{Z}$),
- Parametric transformations $\mathbf{R}$ and $\mathbf{S}$ that map the condition-specific latent spaces into each other,
- The predicted expression $\hat{\mathbf{Y}}$ for each gene and cell in any condition,
- The predicted differential expression $\mathbf{\Delta}$ for each gene and cell for any contrast that can be constructed from the design matrix,
- For each gene and contrast, a neighborhood of cells and a statistical measure of significance (*P* value).

The last of these will usually be the one end users care about most; the others are useful for diagnostics, quality assessment, visualization and further modeling uses of the data.

The explicit parameterization of the transformations $\mathbf{R}$ and $\mathbf{S}$ means that they can easily be interpolated or extrapolated beyond the observed set of data and inverted from the low-dimensional embedding back to the data space. In this sense, LEMUR is a generative model.

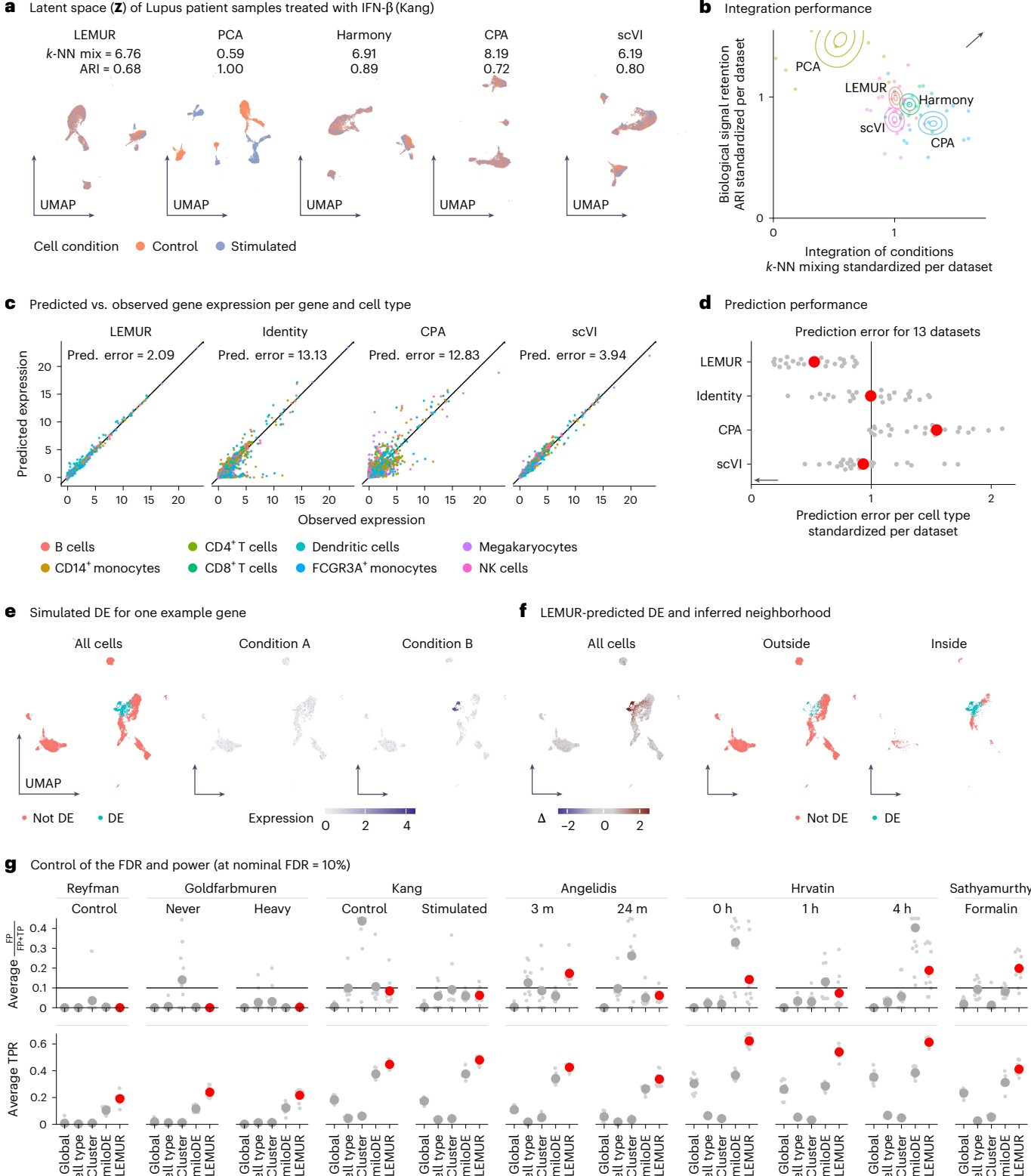

**Fig. 3 | Performance assessment. a,** UMAPs of the latent spaces for the data from Kang et al.[18]. The $k$-NN mixing coefficient and the ARI are defined in the main text. **b,** Density plots of the bootstrapped mean performance for ARI and $k$-NN mixing across 13 datasets. To adjust for dataset-dependent variation, we divided the $k$-NN and ARI scores by the average per dataset. **c,** Scatterplots of predicted (pred.) expression under treatment ($y$ axis) against observed expression ($x$ axis) for 500 genes in each of eight cell types (same data as in **a**). **d,** Prediction error as in **c**, across the same 13 datasets as in **b** (gray points). Red points show the mean. **e,** Simulation setup. Left, for one of the implanted genes, a UMAP is shown of the

LEMUR latent space, where the color indicates whether an expression change in this gene was simulated for that cell. Center and right, simulated expression values. **f,** Left, predicted log fold change for the gene from **e** ($\Delta = \hat{Y}^B - \hat{Y}^A$). Center and right, the set of cells inside or outside the inferred neighborhood. **g,** Comparison of observed false discovery proportion and true positive rate (TPR) for 11 datasets, with ten replicates and the overall mean shown as a large point. The nominal FDR was fixed to 10%. FP, false positive; IFN, interferon; m, months; NK, natural killer; TP, true positive.

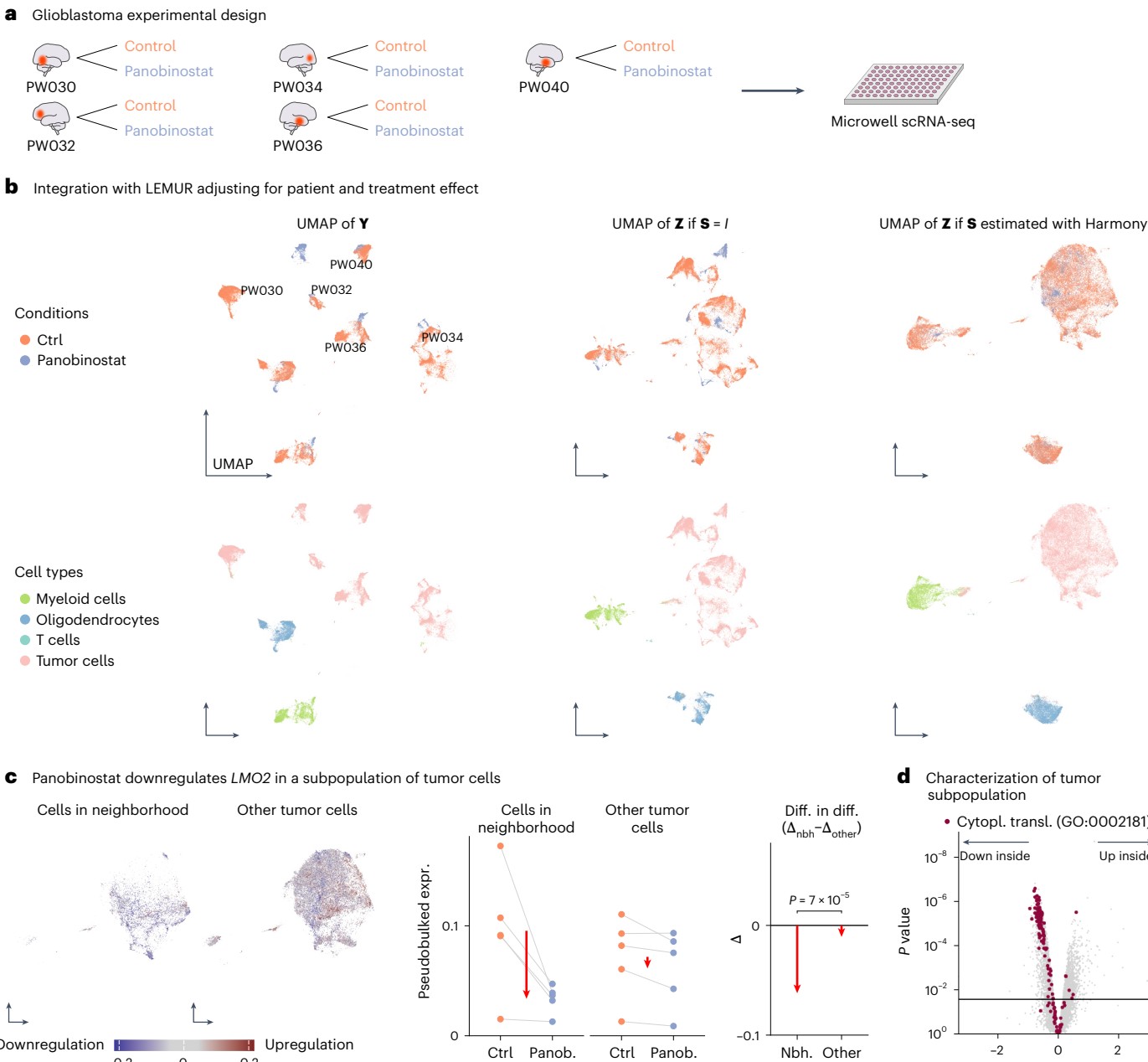

**Fig. 4 | Analysis of five glioblastoma tumor biopsies. a**, Experimental design. **b**, UMAP of the log-transformed data (first column) and the latent embeddings produced by LEMUR (second and third columns), colored by treatment and cell type (first and second row, respectively). **c**, Differential expression analysis of *LMO2* within tumor cells. Faceted UMAP of cells inside and outside the neighborhood. The scatterplots in the middle show pseudobulked expression values from cells inside and outside the neighborhood by donor and condition. Right, comparison of differences inside and outside the neighborhood (red arrows). The *P* value is based on a two-sided likelihood ratio test from a negative binomial count model. **d**, Volcano plot for the comparison between cells inside and outside the subpopulation from f, all in the control condition. Genes annotated as involved with translational activity are highlighted in red. The set of genes above the horizontal line has an FDR of 10%. Nbh., neighborhood; panob., panobinostat; diff. in diff., difference in difference; expr., expression; cytopl. transl., cytoplasmic translation; scRNA-seq, single-cell RNA-seq.

## Performance assessment

We assessed performance of LEMUR using 13 publicly available multi-condition single-cell datasets (listed in Data availability). We preprocessed each dataset consistently following the method of Ahlmann-Eltze and Huber[11].

First, we considered integration performance: how well does the joint low-dimensional representation of the cells preserve biological signal encoded in the latent space but remove traces of the known covariates such as batch and treatment effects? Figure 3a illustrates this on

the dataset from Kang et al.[18] of Lupus patient samples treated with either interferon β or vehicle control. We measured covariate removal by counting for each cell how many of its *k* = 20 neighbors come from the same condition (*k*-nearest neighbor (*k*-NN) mixing). For a balanced dataset with two conditions, an ideal method scores a k-NN mixing value of *k*/2 = 10. We measured the biological signal retention by comparing, for each condition separately, a clustering of the embedding with a clustering of the original data, as measured by the mean of the two adjusted Rand indexes (ARI). An ideal method scores close to ARI = 1.

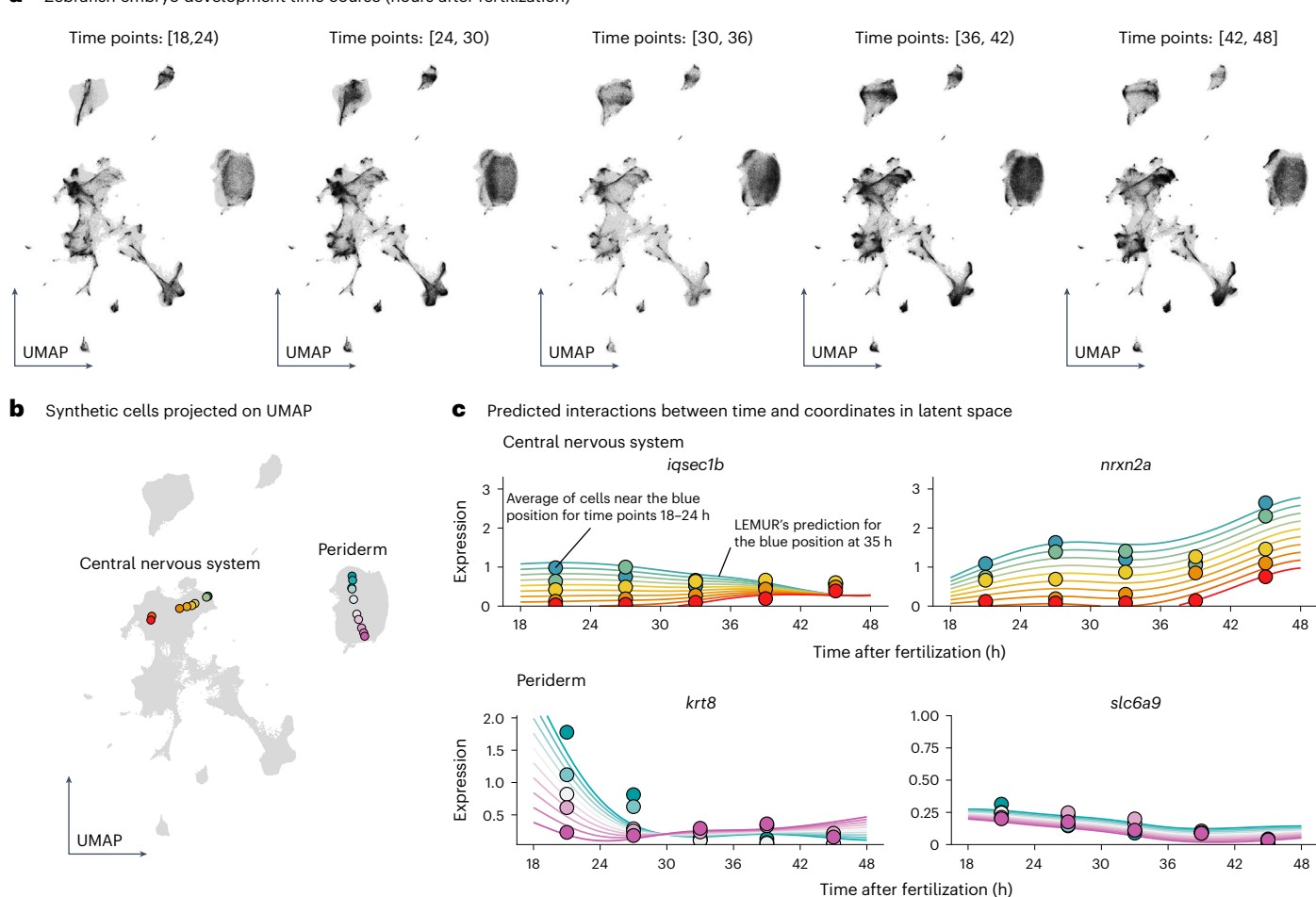

**Fig. 5 | Analysis of a time course single-cell experiment. a**, UMAPs of embryonic development based on the integrated latent space of LEMUR. Black, cells from the respective time window after fertilization; gray, all other cells, for comparison. Because some cell types only exist at particular stages of development, temporal changes in the distribution of black points are to be expected. **b**, Synthetic cells projected onto the UMAP. They interpolate between pairs of observed cells, one pair in the periderm (purple and turquoise) and one in the central nervous system (red and green). **c**, Expression predictions (smooth lines) and averaged observed data (points) for four genes as a function of time (*x* axis) and latent space coordinates (color).

Across the 13 datasets, the performance of LEMUR on these measures was similar to that of Harmony (Fig. 3b). Other methods make different tradeoffs between the two measures, and no method clearly dominates. Extended Data Fig. 1 shows that the results are consistent across seven additional metrics.

The computational cost of running LEMUR is at the low end of what may be expected for such data and comparable to that of other approximative PCA methods. For instance, computing the first 50 latent dimensions on the Goldfarbmuren data, with 24,178 cells and 20,953 genes (which occupies 4 GB of RAM), took us 35 s and 24 GB of RAM with the approximative irlba algorithm[19]. For comparison, fitting LEMUR without integration took 103 s and needed 33 GB of RAM. Aligning the cells with landmarks or Harmony added 2 and 95 s, respectively (Extended Data Fig. 2).

Next, we assessed the ability of LEMUR to predict gene expression across conditions. We used it to predict gene expression under treatment for cells that were observed in the control condition and compared these predictions to data from cells that, in fact, were treated. To avoid overfitting, we assessed predictions on 'held-out' cells, the data of which were not used for training. As there is no direct correspondence between individual cells observed under the two conditions, we considered averages across annotated cell types. Fig. 3c shows scatterplots of the predicted–observed

comparison for the Kang et al.[18] dataset for the 500 most variable genes in eight cell types for four methods: CPA, scVI, LEMUR and the trivial prediction of no change (identity). Across the 13 datasets, LEMUR showed the smallest prediction error measured by the $L_2$ distance between observed and predicted values (Fig. 3d). Extended Data Fig. 3 shows that the results are consistent across six additional metrics.

In a third set of comparisons, we tested the ability of LEMUR to identify sets of cells with consistent differential expression. We took all cells from the control condition of the Kang et al.[18] data, assigned them randomly to a condition A or B and implanted genes with differential expression in a subset of cells. Fig. 3e shows an example. LEMUR accurately identified the expression change and inferred a neighborhood of cells that overlapped well with the simulated ground truth (Fig. 3f).

We expanded this analysis to ten more semi-synthetic datasets, each with 200 implanted differentially expressed genes, to assess the type I error control of the LEMUR differential expression test. LEMUR on average controlled the false discovery rate (FDR) (Fig. 3g, top). In addition, it was more powerful than a pseudobulked test across all cells (global) or separate tests for subsets of cells, either by cell type or cluster as in Crowell et al.[9] or by neighborhood as in miloDE[10] (Fig. 3g, bottom). Extended Data Fig. 4 assesses FDR control and power for additional variants of the LEMUR method. It shows that

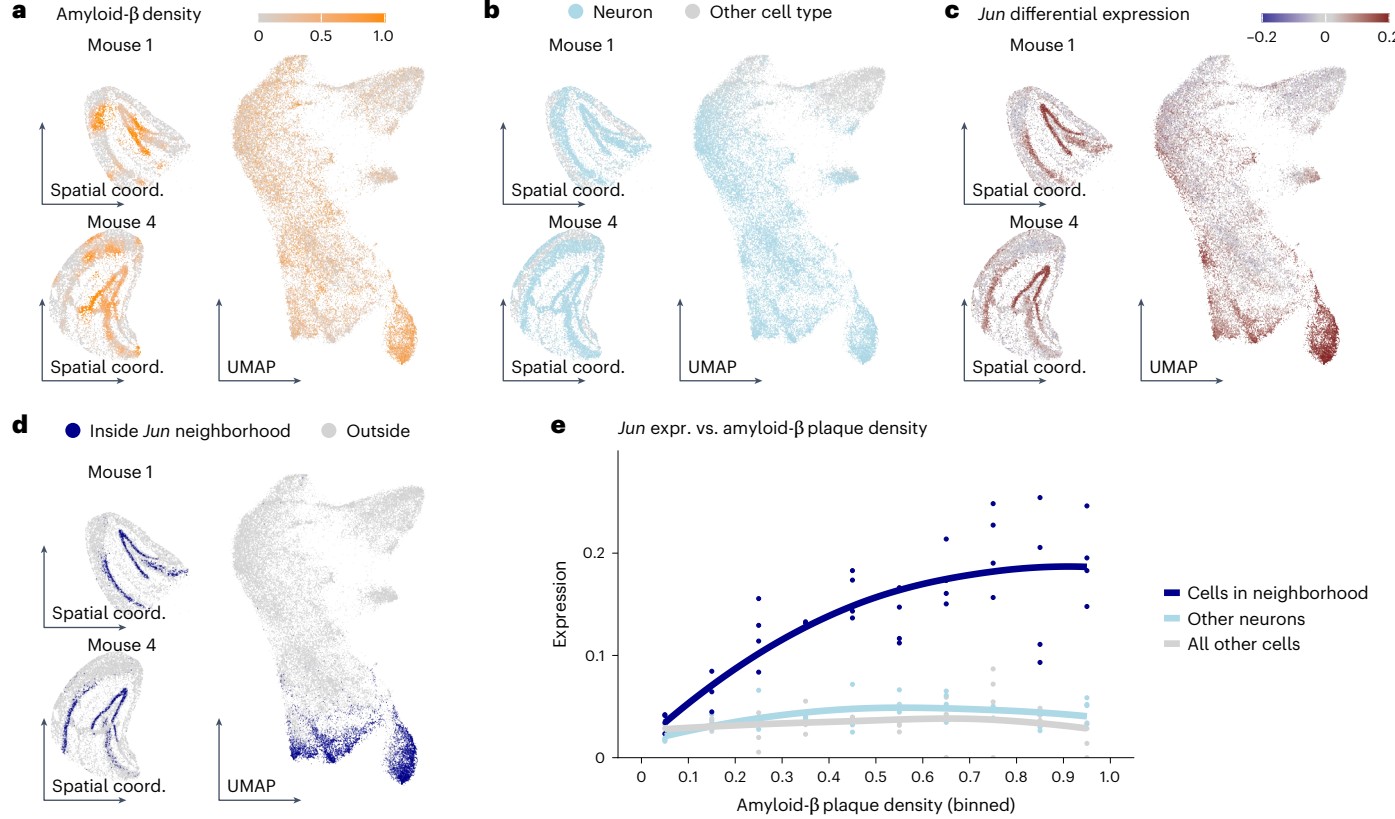

**Fig. 6 | Analysis of a spatial single-cell experiment. a–d**, UMAPs of cells from the hippocampi of four mice and spatial maps for two of them, colored by amyloid-β plaque density (calculated from smoothed and normalized fluorescent images[26]) (**a**), whether the cell is a neuron (**b**), LEMUR-predicted differential expression (log$_2$ fold change) for *Jun* (**c**) and whether the cell is inside the differential expression neighborhood for *Jun* (**d**). **e**, Scatterplot of *Jun* expression as a function of plaque density for cells inside the neighborhood, neurons not in the neighborhood and all other cells. The line is a local regression (LOESS) fit through the pseudobulked values. Coord., coordinate.

accounting for post-selection bias from the neighborhood inference is important and that it is successfully addressed by data splitting.

Overall, these benchmarks demonstrate that LEMUR (1) successfully integrates single-cell data from different conditions, (2) detects cell type- and state-specific differential expression patterns without access to prior clusterings or categorizations of cells and (3) provides accurate statistical type I error control and good power. In the following, we apply LEMUR to the analysis of different biological datasets.

**Treatment–control data: panobinostat in glioblastoma**

Zhao et al.[20] reported single-cell RNA-seq data of glioblastoma biopsies. Aliquots from five patients were assayed in two conditions: control and panobinostat, a histone deacetylase (HDAC) inhibitor (Fig. 4a). After quality control, the data contained 65,955 cells (Extended Data Table 1).

The left column of Fig. 4b shows a two-dimensional UMAP of the input data **Y**. Most visible variation is associated with the known covariates: donor and treatment condition. Some further variation is related to the different cell types in the biopsies. We used LEMUR to absorb donor and treatment effects into **R**, setting the latent space dimension to *P* = 60. The middle column of Fig. 4b shows, upon fixing **S**(*x*) to the identity matrix, a UMAP of the resulting matrix **Z** of latent coordinates for each cell. Cells from different samples are more intermixed, and within-sample cellular heterogeneity is more evident. This picture becomes even clearer after using **S** to match cell subpopulations across samples using Harmony's maximum diversity clustering (Fig. 4b, right column). Here, a large population of tumor cells and three non-tumor subpopulations become apparent.

The successive improvement of the latent space representation from left to right is further demonstrated in the bottom row of Fig. 4b, where the points are colored according to a cell type assignment that we obtained from the expression of selected marker genes and known chromosomal aneuploidies (Extended Data Fig. 5a,b).

The linear latent space of LEMUR is readily interpretable. This is exemplified in Extended Data Fig. 5c, which extends the biplot concept from PCA[21] to the multi-condition setting. We can explore how higher or lower expression of any gene affects a cell's position in the latent space **Z** by plotting the gene's loading vector relative to the coordinate system of **Z**.

Using LEMUR's differential expression testing, we found that panobinostat caused cell subset-specific expression changes in 25% of all genes (2,498 of 10,000) at an FDR of 10%. Extended Data Fig. 6 shows the differential expression and inferred neighborhoods for seven genes.

Focusing on tumor cells, we identified subpopulations that differentially responded to panobinostat treatment (Fig. 4c). In a subpopulation of 9,430 tumor cells, which stemmed in substantial proportions from all five patients, treatment with panobinostat caused downregulation of *LMO2*, while, in the majority of tumor cells (*n* = 36,535), expression of *LMO2* was unchanged (Fig. 4c, right). The product of *LMO2* forms protein complexes with the transcription factors TAL1, TCF3 and GATA; it is important for angiogenesis and was originally discovered as an oncogene in T cell acute lymphoblastic leukemia[22]. Kim et al.[23] studied the role of *LMO2* in glioblastoma and found that higher expression is associated with worse patient survival. They concluded that *LMO2* could be a clinically relevant drug target. To further characterize the subset of tumor cells that respond to panobinostat

by downregulating *LMO2*, we compared their overall gene expression profiles in the control condition to that of the other tumor cells. We found lower expression of ribosomal genes, consistent with lower translational activity (Fig. 4d).

## Time course data: zebrafish embryo development

Saunders et al.[24] reported an atlas of zebrafish embryo development, which includes data from 967 embryos and 838,036 cells collected at 16 time points in 2-h intervals from 18 h to 48 h after fertilization (Fig. 5a). They used their single-cell RNA-seq data to assign each cell to a common cell type classification scheme and studied the temporal dynamics of appearance and disappearance of cell types along developmental time. We asked whether gene expression changes could reveal additional biological phenomena. Thus, we looked for temporal profiles of gene expression that systematically differed across cells that shared the same cell type annotation. For this, we used the ability of LEMUR to predict any gene's expression at any point in latent space at any time. To represent the time dependence of each gene with a smaller number of parameters, we used natural cubic splines with three degrees of freedom, following the method of Smyth et al.[25]. To model latent space variation, we interpolated linearly in latent space **Z** between two cells (red and green) from the central nervous system and, analogously, between two cells (purple and turquoise) from the periderm, a transient, outermost epithelial layer that covers the developing embryo during the early stages of development (Fig. 5b and Extended Data Fig. 7a).

We used the LEMUR fits to screen for genes for which the spline coefficients were different across this interpolated latent space gradient, according to a statistical test for interaction (FDR = 0.001). Fig. 5c shows the data for four examples in which the temporal divergence of gene expression was corroborated by pseudobulking the observed expression data of the nearest neighbors in each 6-h interval. For instance, *krt8*, which encodes a keratin essential for the structural integrity, protective barrier function and proper development of the periderm, showed decreasing expression over time for cells close to the turquoise cell and increasing expression for cells close to the purple cell. Several possible explanations for the intricate and divergent temporal regulation of *krt8* within periderm cells (Extended Data Fig. 7b,c) exist, including spatial structures.

## Spatial data: plaque density in Alzheimer's disease

Cable et al.[26] performed Slide-seqV2 on the hippocampus of four mice genetically engineered to model amyloidosis in Alzheimer's disease. Using microscopy, they quantified the spatial density of amyloid-β plaques (Fig. 6a). Thus, plaque density is an observational covariate that varies from cell to cell; this is in contrast to the covariates considered above, which vary from sample to sample. Cable et al.[26] reported a differential expression analysis per discrete cell type category; however, the categories were fairly broad and could not account for the gradual changes suggested by the data (Fig. 6b and Extended Data Fig. 7a). LEMUR enabled us to defer cell type categorization and directly identify genes with expression varying between low and high plaque density in adaptively found subsets of cells. Fig. 6c shows the differential expression prediction for *Jun*, encoding a transcription factor that was identified as a member of the pathway regulating β-amyloid-induced apoptosis[27,28] and one of the top hits after the LEMUR analysis. The correlation between higher amyloid-β plaque density and increased *Jun* expression was limited to a subset of about 20% of the neurons, which clustered both in spatial coordinates and in the latent space (Fig. 6d). The correlation did not hold for other neurons (Fig. 6e and Extended Data Fig. 8b). We projected the data from Cable et al. onto the hippocampus reference atlas of Yao et al.[29] and found that this subset belonged to glutamatergic neurons from the dentate gyrus and CA1 (prosubiculum) (Extended Data Fig. 8a).

## Discussion

LEMUR enables differential expression analysis of single-cell-resolution expression data between composite samples, such as tissue biopsies, organs, organoids or whole organisms. The method allows for arbitrary experimental or study designs, specified by a design matrix, just as in ordinary linear regression or in omics-oriented regression methods like limma[13], edgeR[17] and DESeq2 (ref. 14). Applications range from comparisons between two conditions with replicates, over paired studies, such as a series of tissue biopsies before and after treatment, over studies with multiple covariates (for example, genetic and drug perturbations), interactions between covariates and continuous covariates. The method represents cell-to-cell variation within a condition using a continuous latent space. Thus, it avoids, or postpones, the need for categorical assignment of cells to discrete cell types or cell states and offers a solution to one of the challenges identified by Lähnemann et al.[1].

We demonstrated the utility of the approach on three prototypical applications, and we benchmarked important aspects of its performance on a compendium of 13 datasets. The application cases are a matched control–treatment study of patient samples in glioblastoma, an atlas of zebrafish embryo development in which time is a continuous covariate and a spatial transcriptomic study of Alzheimer's plaques in which plaque density is a continuous covariate. We showed how LEMUR identified biologically relevant cell subpopulations and gene expression patterns.

To achieve this, we combine latent space representation by dimension reduction with regression analysis in a new matrix factorization approach. The model is predictive: for each observed cell, it predicts its gene expression in any of the conditions, even though it was only measured exactly once. Moreover, as each cell is parameterized by a position in $P$-dimensional real vector space, the model can also predict the expression of 'synthetic cells' at unobserved positions, for instance, in between observed cells or extrapolating out, in any condition. We use these capabilities for differential expression analysis.

We detect neighborhoods of cells with consistent differential expression patterns with respect to comparisons ('contrasts') of interest. The neighborhoods are found in a data-driven manner. No a priori categorization of cells into 'cell types' is needed, but, once neighborhoods have been identified, one can annotate or compare them with whatever annotation that is relevant.

Our current implementation of neighborhood finding leaves room for future improvements. It is stochastic, by relying on a random sample of one-dimensional projections of point clouds in $P$-dimensional space. Thus, repeated running of the algorithm can result in (slightly) different outputs. Also, it addresses the post-selection inference problem using a rather heavy-handed data-splitting approach.

Unlike some other single-cell data integration and expression prediction tools, LEMUR is built around linear methods. It is parameterized with a modest number of parameters and uses a small number of layers. This is in contrast to the often-repeated claim that the complicatedness of single-cell data necessitates nonlinear methods and 'deep' models. We showed that our approach based on simple, linear matrix decomposition using a sufficiently high-dimensional latent space is capable of representing the data in a useful manner. Compared to deep-learning approaches, LEMUR's interpretable and easy-to-inspect model should facilitate dissection and follow-up investigation of its discoveries. In Supplementary Note 1, we discuss how the LEMUR model differs from other approaches that combine dimension reduction and regression.

Overall, we believe that LEMUR is a valuable tool for first-line analysis of multi-condition single-cell data. Compared to approaches that require discretization into clusters or groups before differential expression analysis, representing cell types and states in a continuous latent space may better fit the underlying biology and may facilitate the precise identification of affected cells. This in turn should ease analysts' work and enable biological discoveries that could otherwise be missed.

## Online content

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

## Methods

Our study complies with all relevant ethical regulations. A specific ethics board approval was not needed, as we developed a new computational method and reanalyzed publicly available data.

Given the data matrix $\mathbf{Y}$ of size $G \times C$, where $G$ is the number of genes and $C$ is the number of cells, consider the decomposition

$$\mathbf{Y} = \mathbf{R}\,\mathbf{Z} + \gamma_{\text{offset}} + \boldsymbol{\varepsilon}, \tag{1}$$

with the $G \times P$-dimensional matrix $\mathbf{R}$, the $P \times C$ matrix $\mathbf{Z}$, the $G \times C$ matrix $\boldsymbol{\varepsilon}$, the $G$-dimensional vector $\gamma_{\text{offset}}$ (all real valued) and $P < \min(G, C)$. Note that we use a recycling convention like the one used in the R language for the sum operator (+) for a matrix A and a conformable column vector $b$: $(A + b)_{ij} = A_{ij} + b_i$. To simplify interpretation, we require the columns of $\mathbf{R}$ to be orthonormal, that is, they form an orthonormal basis of a P-dimensional linear subspace of $\mathbb{R}^G$. $\mathbf{Z}$ can then be considered the coordinates of $C$ points in that linear subspace, each representing a cell. We call it a P-dimensional embedding of the cells. Setting $\gamma_{\text{offset}}$ to the row-wise means of $\mathbf{Y}$, the matrices $\mathbf{Z}$ and $\mathbf{R}$ are fit by minimizing the sum of squared residuals

$$\sum_{g=1}^{G} \sum_{c=1}^{C} \boldsymbol{\varepsilon}_{gc}^2. \tag{2}$$

PCA is a special case in which, in addition, the columns of $\mathbf{R}$ are obtained from an eigendecomposition and ordered by eigenvalue. Alternatively, PCA can be understood as the decomposition in which also $\mathbf{Z}$ is orthogonal, which emphasizes the relation to singular-value decomposition. In the applications considered in this work, $P \ll \min(G, C)$, and $\mathbf{R}$ and $\mathbf{Z}$ can be considered a lower-dimensional approximation of the full data matrix $\mathbf{Y} - \gamma_{\text{offset}}$.

We extend equation (1) to incorporate known covariates for each cell. Thus, we consider not just a single matrix $\mathbf{R}$ and a single vector $\gamma_{\text{offset}}$ but treat them as functions of the covariates,

$$\begin{aligned} \mathbf{R} &: \mathbb{R}^K \rightarrow \{\mathbf{A} \in \mathbb{R}^{G \times P} | \mathbf{A}^T\mathbf{A} = \mathbf{I}_P\} \\ \gamma &: \mathbb{R}^K \rightarrow \mathbb{R}^G, \end{aligned} \tag{3}$$

where the arguments of these functions are rows of the $C \times K$ design matrix $\mathbf{X}$, that is, elements of $\mathbb{R}^K$. The range of the function $\mathbf{R}$ is the set of orthonormal $G \times P$ matrices whose Gramian $\mathbf{A}^T\mathbf{A}$ is the P-dimensional identity matrix $\mathbf{I}_P$. Equation (1) then becomes, for each cell $c$,

$$\mathbf{Y}_{:c} = \mathbf{R}(\mathbf{X}_{c:})\,\mathbf{Z}_{:c} + \gamma(\mathbf{X}_{c:}) + \boldsymbol{\varepsilon}_{:c}. \tag{4}$$

Setting $\gamma(\mathbf{X}_c)$ to the least-sum-of-squares solution of regressing $\mathbf{Y}$ on $\mathbf{X}$, the matrix $\mathbf{Z}$ and $\mathbf{R} \in \mathcal{R}$ are fit by minimizing the sum of squared residuals (equation (2)), where $\mathcal{R}$ is a suitable set of matrix-valued functions that we define in the following. Equation (4) with these additional features can be considered a multi-condition extension of PCA.

Intuitively, this multi-condition PCA finds a function $\mathbf{R}$ that generates for each condition (for each distinct row of the design matrix $\mathbf{X}$) a P-dimensional subspace that minimizes the distance to the observed data in that condition by rotating a common base space into the optimal orientation (Fig. 1c, step 1). $\mathbf{Z}$ is the orthogonal projection of the data on the corresponding subspace. Stability is ensured by constraining $\mathbf{R}$ to come from a set $\mathcal{R}$ of 'well-behaved', smooth functions of the covariates.

To construct the function space $\mathcal{R}$, we recall some concepts from differential geometry. Given the whole numbers $G$ and $P$, the set of all orthonormal real matrices of dimension $G \times P$ is a differentiable manifold, the Stiefel manifold $V_P(\mathbb{R}^G)$. For our application, it is appropriate to consider two matrices equivalent if they span the same linear subspace of $\mathbb{R}^G$. The set of all such equivalence classes is again a differentiable manifold, called the Grassmann manifold $\text{Gr}(G, P)$[30,31]. Accordingly,

elements of $\text{Gr}(G, P)$ are P-dimensional linear subspaces of the ambient space $\mathbb{R}^G$. Computationally, we represent an element of $\text{Gr}(G, P)$ by an orthonormal matrix, that is, by one of the members of the equivalence class.

We then construct $\mathcal{R}$ as the set of all functions $\mathbf{R}$ that have domain and codomain as in equation (3) and can be written as

$$\mathbf{R}_\mathbf{B}(x) = \text{Exp}_\mathbf{o}^{(\text{Gr})}\left(\sum_{k=1}^{K} x_k \mathbf{B}_{::k}\right), \tag{5}$$

where $\mathbf{B}$ is a three-dimensional real-valued tensor of size $G \times P \times K$. The expression $\text{Exp}^{(\text{Gr})}$ is the exponential map on the Grassmann manifold. It takes a point $\mathbf{o} \in \text{Gr}(G, P)$ and a tangent vector at that point and returns a new point on the Grassmann manifold. Thus, given a choice of the point $\mathbf{o}$, which we call the base point, and of the design matrix $\mathbf{X}$, the set of all possible $\mathbf{B}$ induces $\mathcal{R}$, and fitting equation (4) is achieved by fitting $\mathbf{B}$. We use the terms tangent vector and tangent space in their standard meaning in differential geometry and represent tangent vectors with $G \times P$ matrices. The name exponential map derives from the fact that, for some Riemannian manifolds, the exponential map coincides with the matrix exponential; however, this is not the case for Grassmann manifolds. Here, the exponential map for a base point $\mathbf{o}$ and a tangent vector represented by $\mathbf{A} \in \mathbb{R}^{G \times P}$ is

$$\text{Exp}_\mathbf{o}^{(\text{Gr})}(\mathbf{A}) = \mathbf{o}\,\mathbf{V}\,\text{diag}(\cos d)\,\mathbf{V}^T + \mathbf{U}\,\text{diag}(\sin d)\,\mathbf{V}^T, \tag{6}$$

where $\mathbf{U}$ and $\mathbf{V}$ are the left and right singular matrices of the singular-value decomposition of $\mathbf{A}$ ($\mathbf{A} = \mathbf{U}\,\text{diag}(d)\,\mathbf{V}^T$)[30].

The argument of the exponential map in equation (5) is a linear combination of the slices of $\mathbf{B}$. Each slice of $\mathbf{B}_{::k}$ represents a tangent vector ($\mathbf{B}_{::k} \in \mathcal{T}_\mathbf{o}\text{Gr}(G, P)$) and so are their linear combinations (a tangent space is a vector space).

We analogously parameterize the offset function $\gamma(x) = \sum_k \mathbf{\Gamma}_{:k} x_k$, where $\mathbf{\Gamma} \in \mathbb{R}^{G \times K}$. Accordingly, fitting $\gamma$ is just ordinary linear regression.

### Non-distance preserving extension

Multi-condition PCA (equation (4)) fits subspaces that approximate the data for each condition but does not depend on shape, scale or, more generally, the statistical distribution of the cells' embeddings in that subspace. Therefore, it can also not adjust for ('absorb') such differences. This rigidity increases stability and can be a desirable model feature for some applications by preventing overfitting, but, for other applications, it can also be a limitation. We extend equation (4) with an optional term $\mathbf{S}$, a non-distance preserving, affine isomorphism of $\mathbb{R}^P$, to (1) obtain additional flexibility and (2) enable input of prior knowledge and user preferences in cell matching:

$$\mathbf{Y}_{:c} = \mathbf{R}(\mathbf{X}_{c:})\mathbf{S}(\mathbf{X}_{c:})\mathbf{Z}'_{:c} + \gamma'(\mathbf{X}_{c:}) + \boldsymbol{\varepsilon}_{:c}. \tag{7}$$

Here, $\mathbf{Z}'_{:c} := \mathbf{S}^{-1}(\mathbf{X}_{c:})(\mathbf{Z}_{:c} - s_0(\mathbf{X}_{:c}))$. The extra term $\mathbf{S}(x)$ distinguishes equation (7), the LEMUR model, from its special case for $\mathbf{S} \equiv \mathbf{I}_P$, multi-condition PCA, equation (4). To allow translations, we also change $\gamma'(x) = \gamma(x) + \mathbf{R}(\mathbf{X}_{c:})s_0(x)$, with $s_0$ defined below.

Next, we describe the selection of $\mathbf{S}$ and $s_0$. It is designed to enable the analyst to state preferences about which sets of cells from different conditions should be considered to match each other, that is, are intended to be the same. We expect such a specification in the form of $E \in \mathbb{N}_0$ sets $\mathbb{E}_1, \ldots, \mathbb{E}_E$, where each $\mathbb{E}_i \subset \{1, \ldots, C\}$ and $\mathbb{E}_i \cap \mathbb{E}_j = \varnothing$ for $i \neq j$. These can be derived, for example, from a set of matching cell type annotations (landmarks) or Harmony's maximum diversity clustering. This provision of preferences is optional; if it is lacking, $E = 0$, the first term in equation (8) vanishes, the optimization simply results in $\mathbf{S} = \mathbf{I}$, the identity, and LEMUR reverts to multi-condition PCA. $\mathbf{S}$ is obtained as a solution to the optimization problem

$$\underset{\mathbf{S}, s_0 \in \mathcal{S},}{\arg\min} \sum_{e=1}^{E} \sum_{c \in \mathbb{E}_e} \left(\mathbf{M}_e - \mathbf{S}^{-1}\left(\mathbf{X}_{c:}\right) \; \left(\mathbf{Z}_{:c} - s_0\left(\mathbf{X}_{c:}\right)\right)\right)^2 + \lambda \left(\left\|\mathbf{W}^{(0)}\right\|_2^2 + \|\mathbf{W}\|_2^2\right),$$

(8)

where the optimization domain $\mathcal{S}$ is described in the next paragraph and $\mathbf{M}_e = |\#\mathbb{E}_e|^{-1} \sum_{c \in \mathbb{E}_e} \mathbf{Z}_{:c}$ is the mean latent space coordinate of the cells in similarity set $\mathbb{E}_e$.

The optimization domain $\mathcal{S}$, that is, the set of possible $\mathbf{S}(x)$ and $s_0$, is the set of affine transformations

$$\mathbf{S}(x) = \mathbf{I} + \sum_{k=1}^{K} x_k \mathbf{W}_{::k}$$

(9)

$$s_0(x) = \sum_{k=1}^{K} x_k \mathbf{W}_{:k}^{(0)},$$

which is parameterized by the three-tensor $\mathbf{W}$ with dimensions $P \times P \times K$ and the $P \times K$ matrix $\mathbf{W}^{(0)}$.

The parameter $\lambda$ regularizes the optimization and pulls the result toward $\mathbf{S}(x) = \mathbf{I}_P$ and $s_0(x) = 0$.

We provide additional details on how the LEMUR model is implemented in Supplementary Note 1.

## Execution details

**Integration and prediction benchmark.** The integration benchmark measured the ability of methods to adjust for known covariates while retaining the biological structure of the data. The prediction benchmark measured, for the tools that support it, how well they are able to predict the expression of a cell in arbitrary conditions.

For both benchmarks, we used 13 single-cell datasets[18,32–43] that we downloaded from publicly available sources (see Data availability for details).

We preprocessed each dataset using the transformGamPoi package's shifted_log_transform function with the default parameters. We identified the 500 most variable genes and held out 20% randomly chosen cells.

For the integration benchmark, we compared scVI version 1.1.2 (ref. [5]), CPA version 0.8.3 (ref. [7]), Harmony version 1.1.0 (ref. [2]), LEMUR (version 1.1.5) with $S \equiv \mathbf{I}_P$ (multi-condition PCA), LEMUR with R fixed to the principal vectors of $\tilde{Y}$ (parametric Harmony) and the full LEMUR model. For PCA, we used the fast implementation from the irlba package (version 2.3.5.1).

Our software is available as an R package on Bioconductor[44] and as a Python package on PyPI[45].

For the prediction benchmark, we did not consider Harmony, as it does not support going back from the integrated embedding to the gene expression space. Instead, we included two other comparisons, which can be considered baseline controls: a linear model-based method that predicts the mean of each condition and identity prediction, which returns the original expression observed for a cell independent of the requested condition.

We tried to run the methods as much as possible with the default parameters. On the advice of the authors, we ran CPA directly on the counts (that is, not on the variance-stabilized data) with the parameters from the tutorial (https://cpa-tools.readthedocs.io/en/latest/tutorials/Kang.html) on integrating the Kang dataset (most importantly, 64 latent dimensions, a negative binomial loss and a learning rate of 0.0003). For the predictions from CPA, we, again based on the recommendations of the authors, log transformed the predicted counts before comparing them to the variance-stabilized data. For LEMUR, we used 30 latent dimensions and a test fraction of 0%. We ran PCA with 30 latent dimensions.

For the evaluation of the integration results displayed in Fig. 3b, we used the integration performance of the held-out data compared to the training data, except for Harmony, which only integrates its training data and has no concept of integrating further, previously unseen data.

To make the output of the different methods comparable, we brought each embedding to a common scale by subtracting the mean for each latent dimension and dividing by the average cell vector length. On this rescaled input, we calculated nine different metrics. Four metrics were used to assess the adjustment for the known covariates:

- $k$-NN mixing. We identified the $k = 20$ nearest neighbors from the training data for each cell from the held-out data. We then calculated how many of those 20 neighbors were from the same condition as the original cell. We averaged these values across all held-out cells to derive a single metric for each dataset-and-method pair.
- Maximum mean discrepancy (MMD). We calculated the MMD discrepancy with a radial basis function kernel between the held-out and training data after subsampling both to a common number of cells[46]. We calculated the discrepancy using scaling factors between 10 and $10^{-3}$ (50 values, log spaced) and 100, 200 and 500 cells and finally averaged the results to obtain a single metric.
- Wasserstein distance. We calculated the Wasserstein distance between held-out and training cells using the Wasserstein function from the transport package after subsampling to a common number of cells. We averaged the results for 100, 200 and 500 cells.
- Variance explained by condition. We calculated the ratio of residual variance after accounting for known covariates over the total variance of the embedding.

We used these four metrics to assess how well each embedding retained the biological information:

- $k$-NN overlap. We calculated a reference embedding with PCA, scVI and CPA that did not try to integrate the conditions. Next, we compared for each condition the similarity of the nearest neighbors on the integrated and non-integrated embeddings. We used the PCA as a reference for Harmony and LEMUR. CPA did not support fitting an embedding without a condition variable; thus, we randomly assigned each cell to a condition and thus created a perfectly mixed dataset in which no additional integration was needed.
- ARI. We calculated a reference embedding with PCA, scVI and CPA that did not try to integrate the conditions. Next, we compared the similarity of a walktrap clustering on the integrated and non-integrated embeddings. We calculated the ARI to measure the cluster consistency using the ARI function from the aricode package.
- Normalized mutual information (NMI). We followed the same procedure as for the ARI but calculated the NMI using the NMI function.
- Variance explained by cell type. We calculated the ratio of residual variance after accounting for the cell types as annotated in the original data over the total variance of the embedding.

Lastly, we also considered one merged metric that directly contrasts the adjustment for the known covariates and the retention of the biological information.

- Variance explained by condition versus cell type. We calculated the ratio of residual variance after accounting for the known conditions plus the cell types over the residual variance accounting only for the cell types.

For the prediction benchmark, we considered a total of ten metrics. We considered two conditions for each dataset and always calculated the predicted expression for condition B for cells from condition A against the observed expression in condition B and vice versa.

- $L_2$ mean. The $L_2$ distance between the mean prediction against the mean observed expression across the whole data.
- $L_2$ mean per cell type. The $L_2$ distance between the mean prediction against the mean observed expression for each cell type.
- $L_2$ of the standard deviation. The $L_2$ distance between the standard deviation of the prediction against the standard deviation of the observed expression across the whole data.
- $L_2$ of the s.d. per cell type. The $L_2$ distance between the standard deviation of the prediction against the standard deviation of the observed expression for each cell type.
- $R^2$ mean. The correlation between the mean prediction against the mean observed expression across the whole data.
- $R^2$ mean per cell type. The correlation between the mean prediction against the mean observed expression for each cell type.
- $R^2$ of the s.d. The correlation between the standard deviation of the prediction against the standard deviation of the observed expression across the whole data.
- $R^2$ of the s.d. per cell type. The correlation between the standard deviation of the prediction against the standard deviation of the observed expression for each cell type.
- MMD. We calculated the MMD discrepancy between the predicted and observed data after subsampling to a common number of cells with a radial basis function kernel[46]. We calculated the discrepancy using scaling factors between 10 and $10^{-3}$ (50 values, log spaced) and 100, 200 and 500 cells and finally averaged the results to obtain a single metric.
- Wasserstein distance. We calculated the Wasserstein distance between predicted and observed data using the Wasserstein function from the transport package after subsampling to a common number of cells. We averaged the results for 100, 200 and 500 cells.

**Variance-explained comparison.** For all 13 datasets used in the integration and prediction benchmark, we compared the fraction of variance explained by LEMUR and by PCA. To make the results comparable, we manually regressed out the effects of the known covariates. We compared irlba's approximate PCA implementation and the LEMUR model accounting for the known covariates. We fixed the linear_coefficient_estimator = 'zero', as we had already manually removed the linear effects. We measured the elapsed time using R's system.time function and the memory using the GNU time command.

**Differential expression and neighborhood inference benchmark.** For the differential expression benchmark, we took the gene expression values of the top 8,000 highly variable genes from individual conditions and appended 200 simulated genes. We assigned each cell to a synthetic control or treatment condition. This ensured that, for all original genes, there was no real differential expression, whereas, for the 200 simulated genes, we were able to control the number of cells with a differential expression pattern and the log fold change.

We only considered the dataset–condition combinations that had the most independent replicates. We chose two conditions from the Angelidis et al.[32], Goldfarbmuren et al.[37] and Kang et al.[18] data, three conditions from Kang et al.[38] and one from Sathyamurthy et al.[42]. Thus, we had 11 datasets in total.

We then simulated 200 genes for each dataset with a varying number of affected cells, which we repeated ten times per dataset. We first performed $k$-means clustering on the 50-dimensional embedding of the data with either two, three, ten or 20 clusters. Next, for each gene, we chose one of the clusters and fixed the log fold change to 0.5, 1, 2 or 4, respectively (that is, for the smaller clusters, we used a larger effect size). The counts were simulated according to

$$\mathbf{Y}_{gc} = \text{GammaPoisson}(\mu = 2^{\boldsymbol{\eta}_{gc}} \text{sf}_c, \alpha = 0.2)$$

$$\boldsymbol{\eta}_{gc} = \beta_g^{(0)} + \beta_g^{(DE)} x_c^{(\text{is DE})} + \beta_g^{(\text{samp 1})} x_c^{(\text{samp 1})} + \cdots , \quad (10)$$

where $\mu$ is the mean and $\alpha$ is the overdispersion parameter of the Gamma-Poisson distribution, $\beta_g^{(DE)}$ is the log fold change, $x_c^{(\text{is DE})}$ indicates whether cell $c$ is inside the selected cluster, $\text{sf}_c$ is the size factor for cell $c$ calculated on the observed genes and $\beta_g^{(\text{samp 1})} x_c^{(\text{samp 1})}$ simulates a sample-specific effect in which $\beta_g^{(\text{samp } i)}$ is drawn from a normal distribution with a standard deviation of 0.1.

Using the known set of simulated genes (true positive) and original genes (true negatives), we calculate the TPR (fraction of identified true positives) and the false discovery proportion (fraction of false positives among all positives).

Our default settings for LEMUR in the differential expression benchmark were

- 30 latent dimensions,
- a test fraction of 50%,
- directions = 'randomized',
- selection_procedure = 'zscore',
- size_factor_method = 'normed_sum' and
- test_method = 'edgeR'.

In addition, we tested several variations:

- LEMUR with test_method set to glmGamPoi[16] or limma[13],
- LEMUR with eight or 80 latent dimensions,
- LEMUR with a test fraction of 20% and 80%,
- LEMUR with size_factor_method = 'ratio',
- LEMUR with directions = 'contrast' or selection_procedure = 'contrast',
- LEMUR with $\mathbf{S} \equiv \mathbf{I}$ (multi-condition PCA) or $\mathbf{R}$ fixed to the principal vectors of $\bar{\mathbf{Y}}$ (parametric Harmony),
- LEMUR where we reused the training data for testing and
- LEMUR where the test and training data were generated through count splitting[47].

We compared the FDR and TPR from LEMUR against seven alternative methods:

- Global test: a single test with edgeR or glmGamPoi across the full dataset
- Cell type test: one test per cell type (using the annotation from the original data) with edgeR or glmGamPoi
- Cluster test: one test per walktrap cluster on the Harmony integrated data with edgeR or glmGamPoi
- miloDE: one test per Milo neighborhood using edgeR[10] (version 0.0.9000; hash, 8803302d).

For the cell type test, the cluster test and miloDE, which perform more than one test per gene, we considered a group of cells as positive if they contained more than 60% changed cells (at least ten). If none of the cell groups for a gene fulfilled this criterion, the group with the largest fraction of changed cells was considered as positive. A group of cells was considered negative if less than 10% of the cells were changed. If the fraction of changed cells was between 10% and 60%, the status for that group of cells was considered indeterminate, and the group was ignored for the TPR and FDR calculations.

**Glioblastoma analysis.** We downloaded the count data and patient annotations from GSE148842. We transformed the counts using the function shifted_log_transform from the transformGamPoi package and filtered out all genes with a total of less than six counts. We filtered out all cells that did not pass the quality filters from the scuttle package's perCellQCFilters function and also those cells that had less than 800 or more than 12,000 counts.

We assigned the cell types and tumor status following the methods from the original publication[20]. We clustered the Harmony integrated data with walktrap clustering into four clusters. Zhao et al.[20] identified a chromosome 7 duplication and a chromosome 10 deletion in all samples. Accordingly, we assigned the cluster that showed upregulation of genes from chromosome 7 and downregulation of genes from chromosome 10 as the tumor cells. The other three clusters were assigned based on marker genes as shown in Extended Data Fig. 5a.

We ran LEMUR with 60 latent dimensions and a test fraction of 50%; otherwise, we used the same defaults as in the differential expression benchmark. The full dataset consisted of three conditions: control, panobinostat and etoposide. For the analysis, we decided to focus on the contrast between panobinostat and control and do not show any data from etoposide-treated cells.

The color scale of $\Delta$ in Fig. 4c was capped at the 95% quantile of the absolute values, squishing more extreme values to the range. The difference of the difference test in the rightmost panel of Fig. 4c was significant at FDR < 0.1 considering all genes with a significant difference (FDR < 0.1) between control and panobinostat inside the neighborhood.

We identified gene ontology terms related with upregulated and downregulated genes using the clusterProfiler package's enrichGO function on the 200 genes with the smallest $P$ value, comparing them against the universe of genes that passed quality control.

**Zebrafish embryonic development analysis.** The data downloaded from GSE202639 were already quality controlled. We subsetted the full dataset to the control cells (ctrl-inj and ctrl-uninj) for the 16 time points between 18 h and 48 h and the 2,000 most variable genes. We transformed the counts using transformGamPoi package's shifted_log_transform function and fit a natural spline model with three degrees of freedom and 80 latent dimensions. We tested the difference between the 48-h and the 18-h time points using the settings from the differential expression benchmark.

For the interpolation in Fig. 5b, we selected two pairs of cells and linearly interpolated their latent position. The selected cells were not from the same time point, but, as they only served as anchors in the latent space $\mathbf{Z}$, this did not influence the results. We projected ten synthetic cells onto the two-dimensional UMAP. We calculated the mean of the observed expression values from the 50 nearest neighbors to five synthetic cells at interpolation points 0, 0.25, 0.5, 0.75 and 1. We predicted the gene expression according to a spline fit for all ten synthetic cells at 50 time points equally spaced between 18 and 48 h.

To prioritize the genes that we manually inspected, we tested whether a spline model with five degrees of freedom could significantly better explain the observed expression pattern over time than a linear model within a selected cell type.

**Alzheimer plaque spatial analysis.** We downloaded the expression data for the four mouse hippocampi with the Alzheimer plaque densities from the BROAD's single-cell repository (https://singlecell.broad-institute.org/single_cell/study/SCP1663). We subsetted the genes to a common set and filtered out lowly expressed genes (total counts per gene less than 50). We further filtered out cells with more than 20% mitochondrial reads and less than 200 or more than 5,000 total counts.

We fit LEMUR with 30 latent dimensions and a test fraction of 60% on an ordered factor of the plaque density cut into ten equally sized bins. We contrasted the largest bin against the smallest bin using the same settings as in the differential expression benchmark.

**Statistics and reproducibility**
Our benchmark compared LEMUR against several other methods for different use cases in single-cell analysis. For each task, we compared against available state-of-the-art methods. To ensure that our results are robust, we ran the analyses on six to 13 independent single-cell

datasets. We did not use any statistical method to predetermine the sample size. No data were excluded from the analyses. The experiments were not randomized. The investigators were not blinded to allocation during experiments and outcome assessment.

Our benchmark is fully reproducible using the code available on Zenodo[48].

**Reporting summary**
Further information on research design is available in the Nature Portfolio Reporting Summary linked to this article.

## Data availability
All datasets used in this study are publicly available: Angelidis et al.[32] (GSE124872), Aztekin et al.[33] (Bioconductor, https://bioconductor.org/packages/scRNAseq), Bunis et al.[35] (Bioconductor, https://bioconductor.org/packages/scRNAseq), Goldfarbmuren et al.[37] (GSE134174), Hrvatin et al.[38] (GSE102827), Jäkel et al.[39] (GSE118257), Sathyamurthy et al.[42] (GSE103892), Kang et al.[18] (Zenodo, https://zenodo.org/records/4473025), Bhattacherjee et al.[34] (Zenodo, https://zenodo.org/records/4473025), Skinnider et al.[43] (Zenodo, https://zenodo.org/records/4473025), Cano-Gamez et al.[36] (Zenodo, https://zenodo.org/records/5048449), Cano-Gamez et al.[41] (Zenodo, https://zenodo.org/records/5048449), Pijuan-Sala et al.[40] (Bioconductor, https://bioconductor.org/packages/MouseGastrulationData), Zhao et al.[20] (GSE148842), Saunders et al.[24] (GSE202639) and Cable et al.[26] (SCP1663). Source data are provided with this paper.

## Code availability
The lemur R package is available at https://bioconductor.org/packages/lemur, and the code to reproduce the analysis is available at https://github.com/const-ae/lemur-Paper, which we also permanently archived using Zenodo https://doi.org/10.5281/zenodo.12726369 (ref. 48). A Python implementation of the LEMUR model (without the differential testing capabilities) is available at https://github.com/const-ae/pylemur.

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

## Acknowledgements

We thank S. Anders, O. Stegle and J. Zaugg for valuable feedback and discussions and the ERC DECODE team for highlighting the needs of multi-condition single-cell RNA-seq analysis. We thank R. Bergmann for his advice on optimization on manifolds. We thank L. Saunders for early access to and discussion of the zebrafish data. We thank EMBL IT services for infrastructure to perform computations. We thank EMBL Kinderhaus for providing authors with thinking time through excellent childcare. This work has been supported by the EMBL International PhD Programme, by the German Federal Ministry of Education and Research (CompLS project SIMONA under grant agreement no. 031L0263A) and by the European Research Council (Synergy Grant DECODE under grant agreement no. 810296). The funders had no role in study design, data collection and analysis, decision to publish or preparation of the manuscript.

## Author contributions

C.A.-E. and W.H. conceived the ideas for the study and wrote the paper. C.A.-E. wrote the software and performed the computations with feedback from W.H.

## Funding

## Competing interests

The authors declare no competing interests.

## Additional information

**Extended data** is available for this paper at https://doi.org/10.1038/s41588-024-01996-0.

**Correspondence and requests for materials** should be addressed to Constantin Ahlmann-Eltze or Wolfgang Huber.

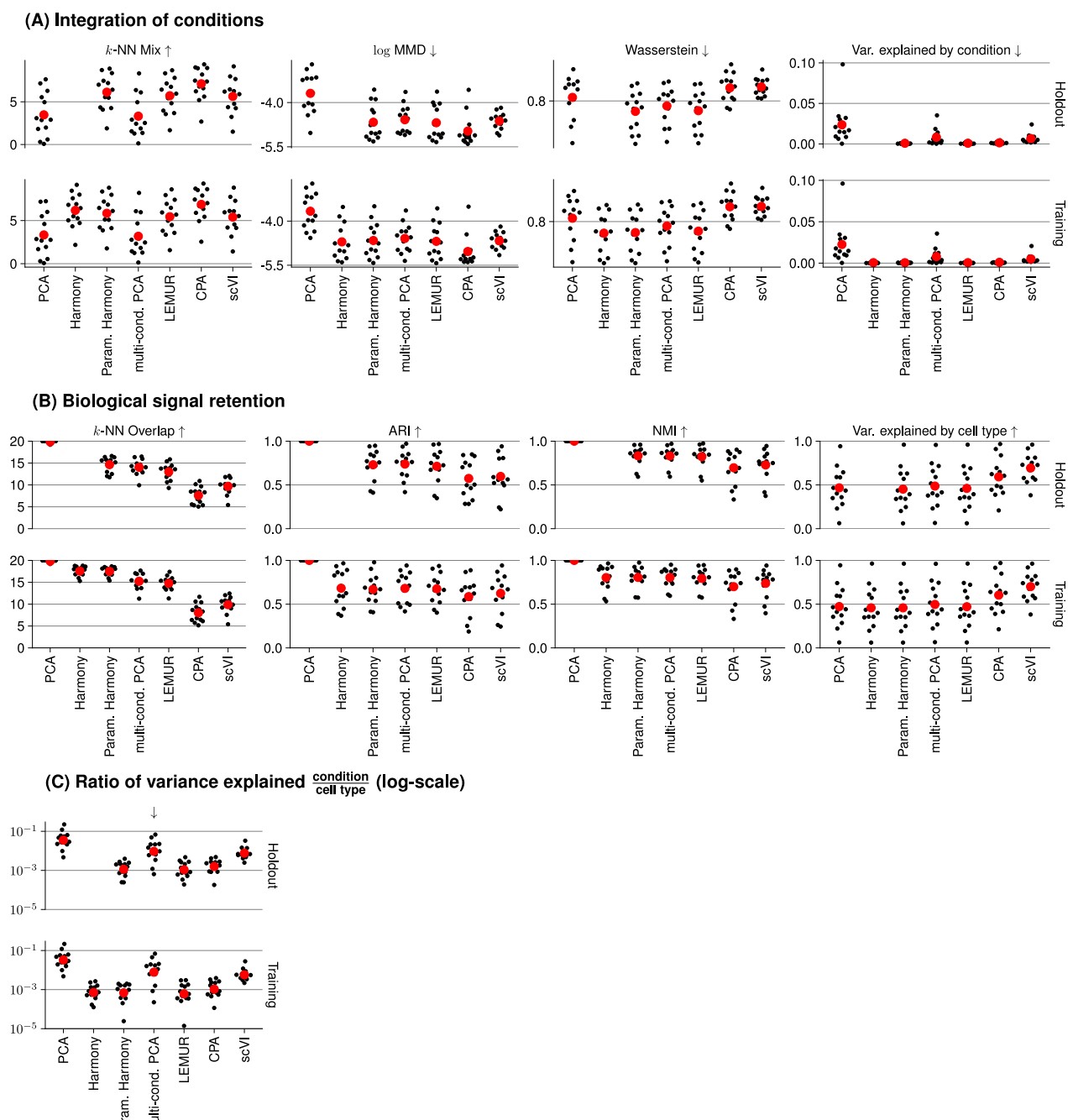

**Extended Data Fig. 1 | Comparison of integration performance across 13 datasets.** (**A**) Beeswarm plots for each metric comparing how well each method adjusted the latent embedding for the known covariates. The arrows next to the metric indicate if higher or lower values indicate better performance. (**B**) Beeswarm plots for each metric comparing how well each method retained the biological signal. (**C**) A beeswarm plot of an integrated performance measure comparing the ratio of variance explained by the known covariates vs cell types (as a proxy for the biological signal). Each black point is the result one dataset and the red points show mean performance. *k-NN*: *k* nearest neighbors, *MMD*: maximum mean discrepancy, *var. expl.*: variance explained, *ARI*: adjusted Rand index, *NMI*: normalized mutual information.

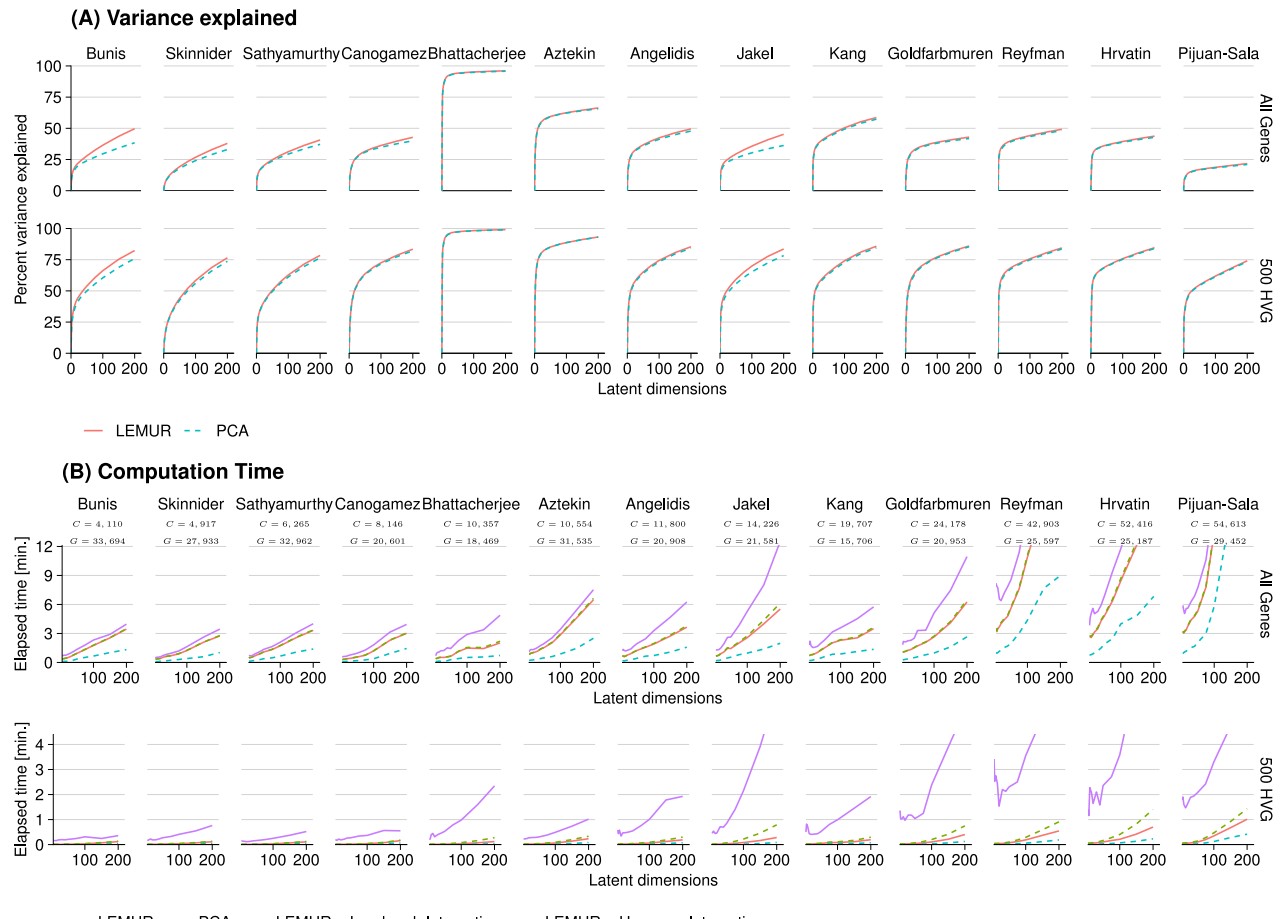

**Extended Data Fig. 2 | Comparison of LEMUR and PCA.** (**A**) Line plot of number of latent dimensions against the variance explained for PCA and LEMUR. (**B**) Line plot of number of latent dimensions against the computation time for PCA, LEMUR, and LEMUR with landmark or Harmony-based integration. The number labelled $C$ and $G$ are the number of cells and genes for each dataset, respectively. *HVG*: highly variable genes.

**Distance and correlation measures between predicted and observed expression**

There are two points per dataset (A vs B and B vs A)

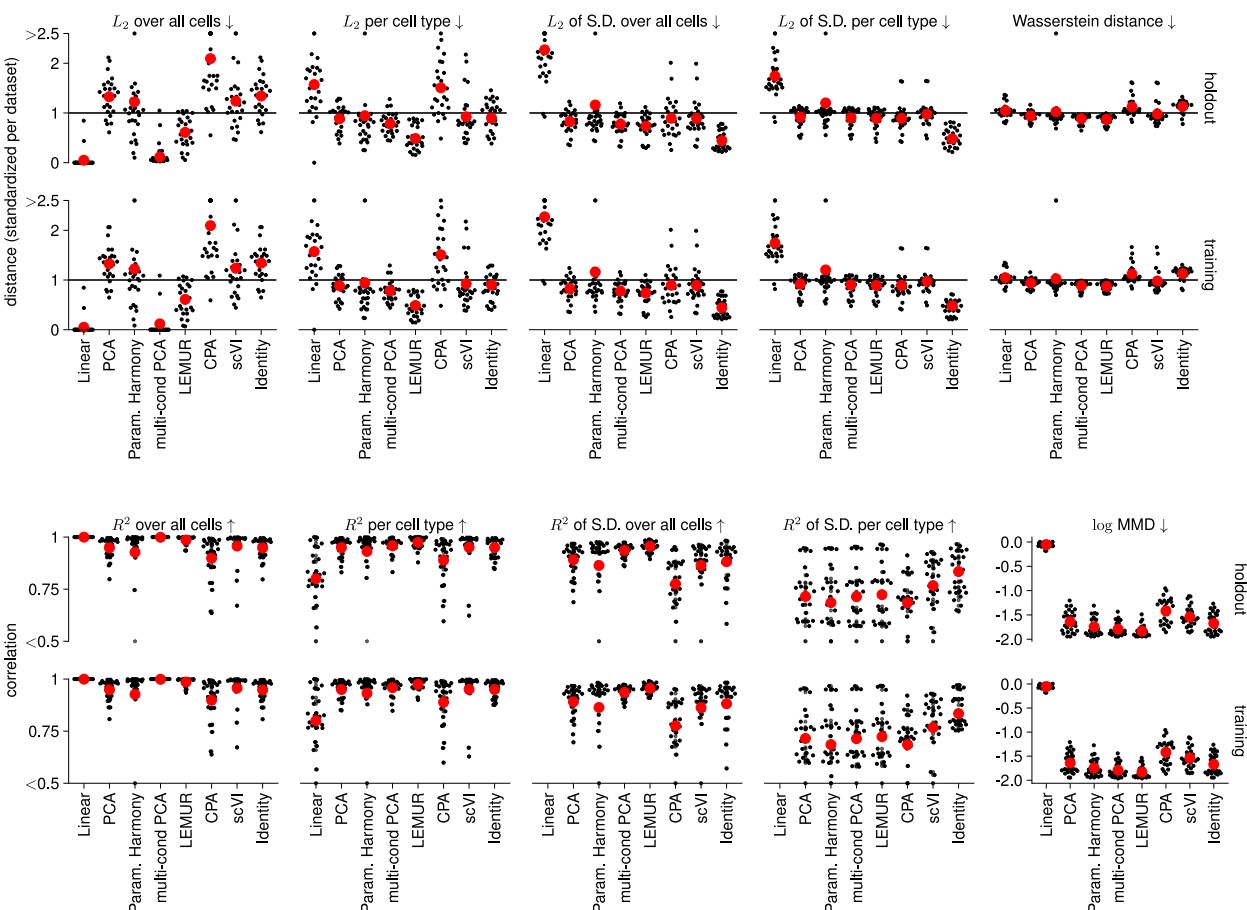

**Extended Data Fig. 3 | Comparison of the ability of each method to predict the expression of genes across conditions for 13 datasets.** The panels show different distance and correlation measures comparing the predicted expression in condition *B* for cells from condition A against the observed expression of cells in condition *B* and *vice versa*. The $L_2$ distance and the correleation where calculated using the mean of the predictions and observations over all cells or per cell type. As the distances varied by two orders of magnitude between datasets, we divided each distance by the mean per dataset. The red points show the mean per method. The arrows next to the metric indicate if larger or smaller values indicate better performance. *S.D.*: standard deviation, *MMD*: maximum mean discrepancy.

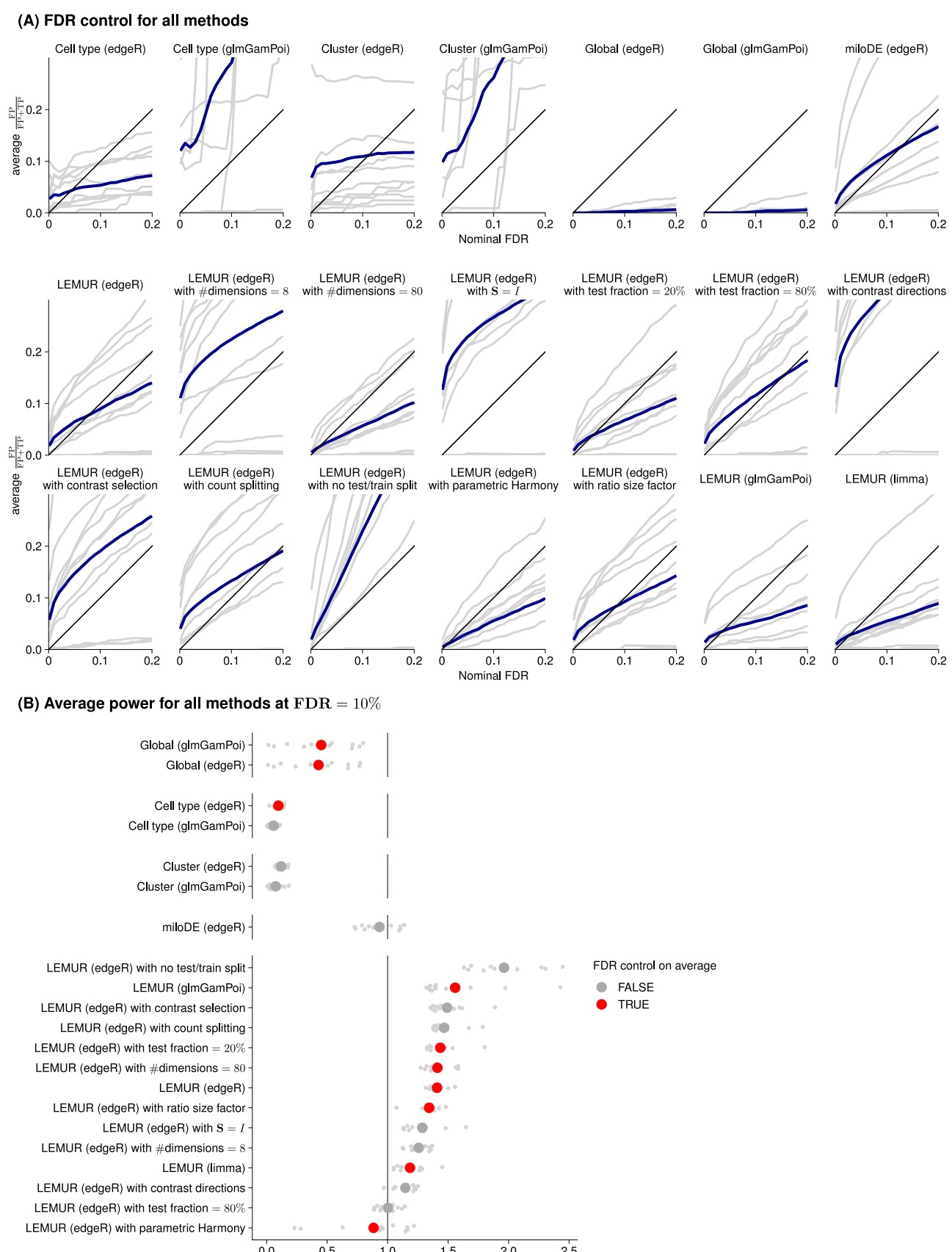

**Extended Data Fig. 4 | FDR control and power for all methods across eleven datasets where we average across the ten replicates per dataset.** (**A**) Line plot of the nominal FDR against the observed false discovery proportion. The FDR is the expectation of the false discovery proportion over many samples. The blue line shows the average. (**B**) Scatter plot of the relative TPR at an FDR = 10% for each method across eleven datasets standardized by the mean per dataset. The point range shows the mean and standard error. If the average observed FDR is larger than 10% the point is greyed out. *FDR*: false discovery rate, *TPR*: true discovery rate.

**(A) Glioblastoma cell type assignment**

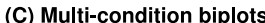

**(B) Aneuploidie**

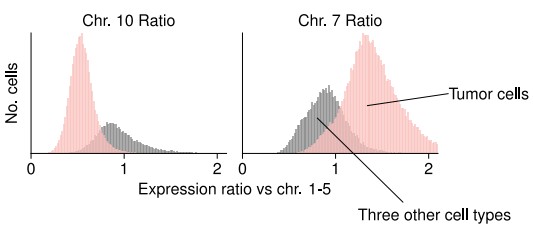

**(C) Multi-condition biplots**

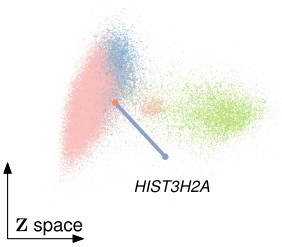

**Extended Data Fig. 5 | Glioblastoma cell type assignment and multi-condition biplots.** (**A**) Marker gene and (**B**) chromosome-aggregated expression levels for each cell type. The the tumor cells are known to bear an amplification of chromosome 7 and a deletion of chromosome 10[20], so we identified the tumor cluster using the ratio of average gene expression on chromosome 7, resp.

10, over the average expression on chromosomes 1-5 (**C**) Multi-condition biplots showing (left) the first two dimensions of the LEMUR latent space (**Z**) for all cells overlayed with arrows representing (middle) three cell type marker genes from Panel A and (right) a gene with large expression change specifically in myeloid cells (*HIST3H2A*, more details in Extended Data Fig. 6).

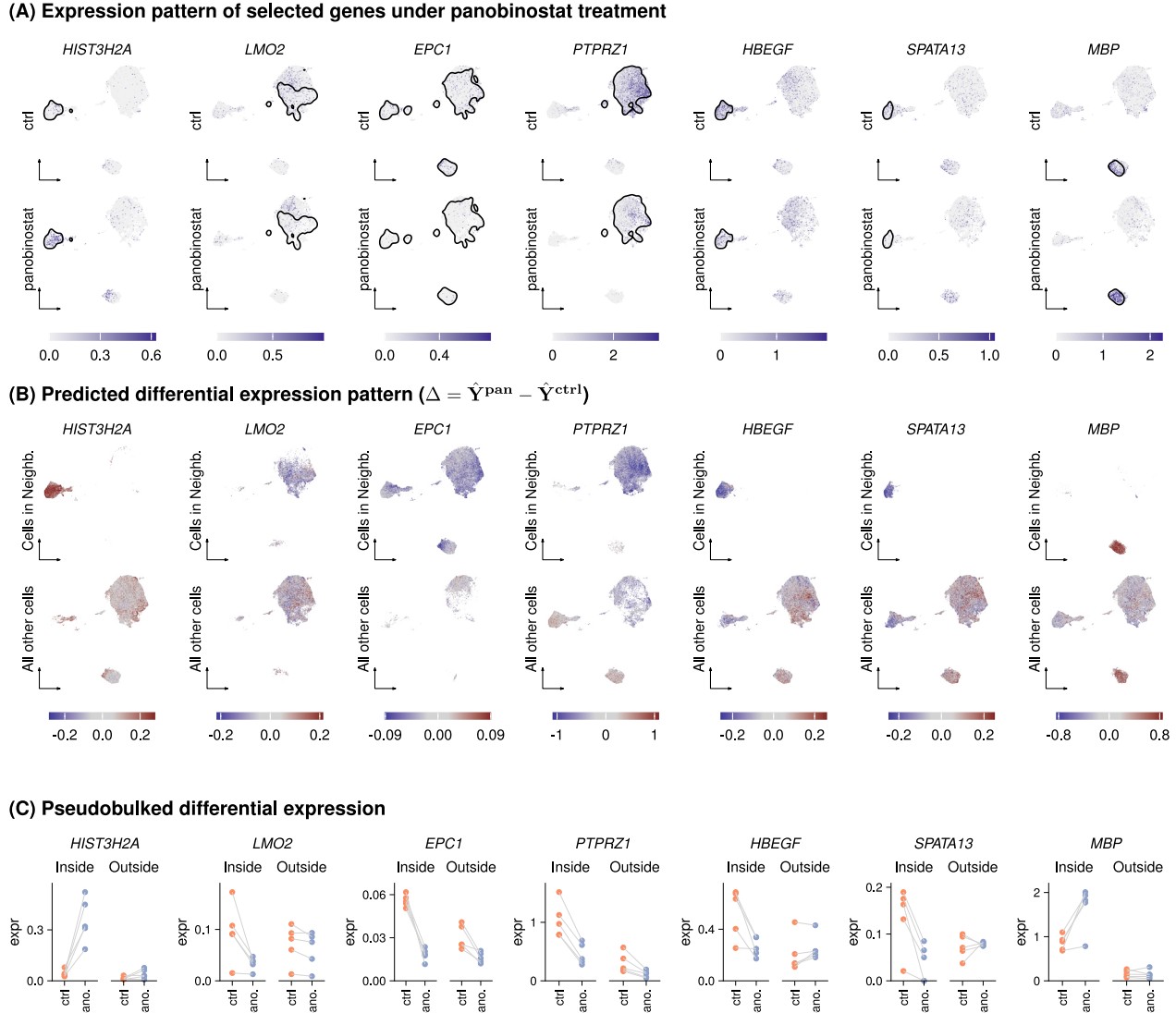

**(A) Expression pattern of selected genes under panobinostat treatment**

**(B) Predicted differential expression pattern ($\Delta = \hat{Y}^{\text{pan}} - \hat{Y}^{\text{ctrl}}$)**

**(C) Pseudobulked differential expression**

**Extended Data Fig. 6 | Differential expression patterns for seven genes with the neighborhoods inferred by LEMUR.** (**A**) UMAPs colored by gene expression (log-normalized counts). The black line encircles 80% of the cells inside the gene-specific differential expression neighborhood. (**B**) UMAPs colored by predicted expression change per cell. The cells are separated depending if they are inside or outside the neighborhood. (**C**) Scatter plot of the pseudobulked expression values per condition, neighborhood status and sample. *ctrl*: control, *neighb.*: neighborhood, *pan*: Panobinostat, *expr*: expression.

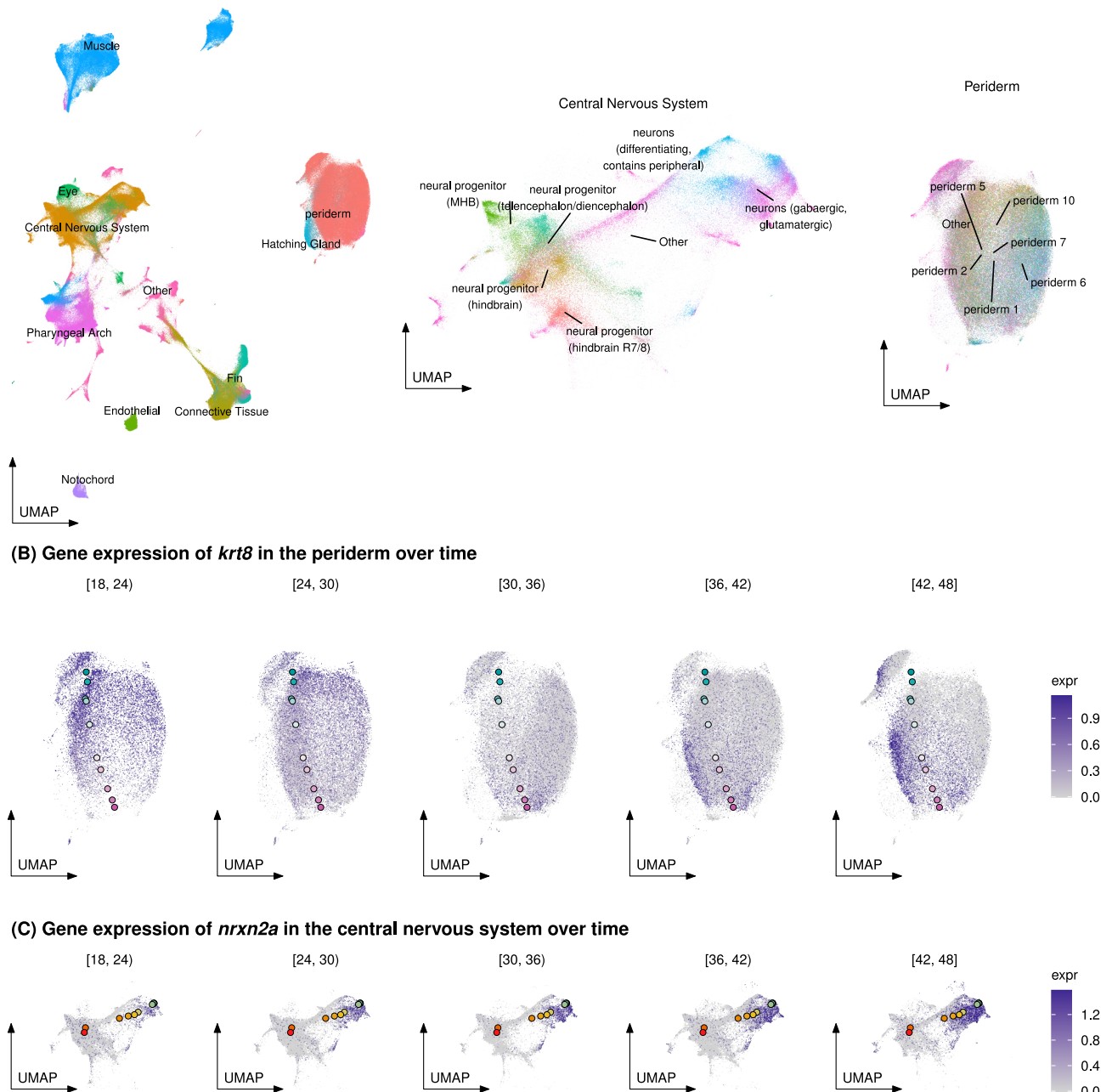

**(A) Zebrafish: tissue labels plus zoom in on cell types in central nervous system and periderm**

**(B) Gene expression of *krt8* in the periderm over time**

**(C) Gene expression of *nrxn2a* in the central nervous system over time**

**Extended Data Fig. 7 | Cell type annotation for zebrafish.** (**A**) Left panel: UMAP of the full timecourse data with cells colored by tissue type as annotated in Saunders et al.[24]. Middle and right panel: Cell type annotations for central nervous system and periderm also from Saunders et al.[24]. (**B**) Gene expression of *krt8* in the periderm over time. (**C**) Gene expression of *nrxn2a* in the central nervous system over time. The overlayed points are the synthetic cells from Fig. 5B.

**(A) Alzheimer's disease mouse model cell type details**

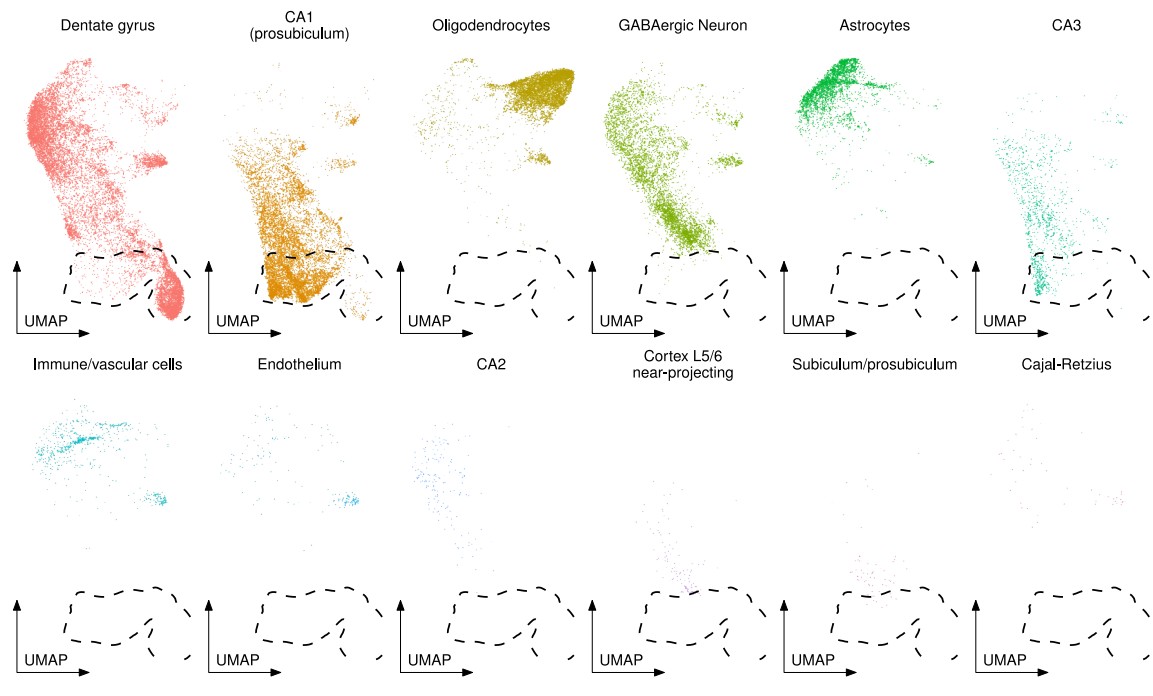

**(B) Relation of Amyloid-$\beta$ density and *Jun* expression per cell type**

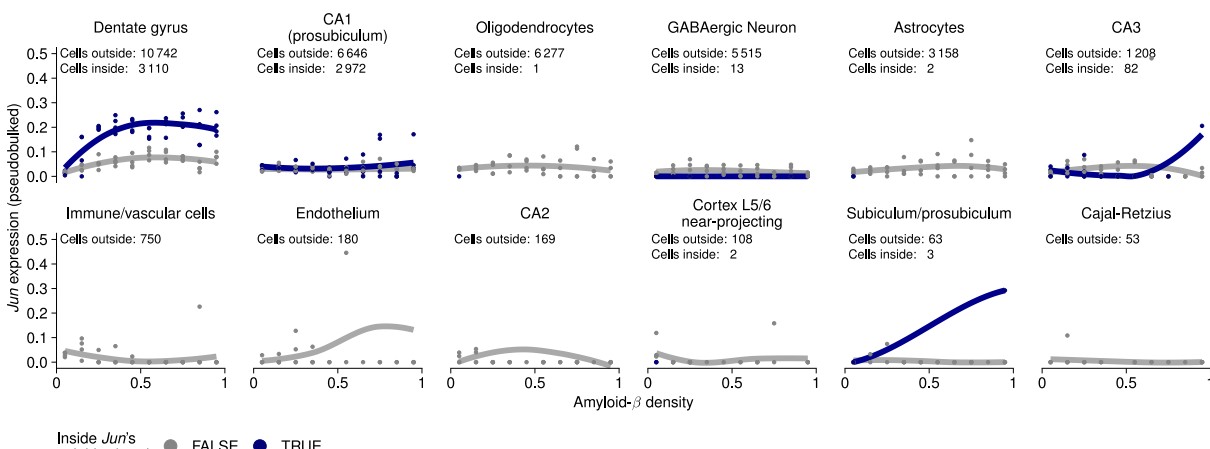

**Extended Data Fig. 8 | Cell type annotation for mouse brain datasets.**
(**A**) UMAP of the Alzheimer's disease mouse model dataset colored and split by cell type annotation provided by Yao et al.[29]. The dashed shape is the 97% density contour of the cells inside the neighborhood. (**B**) Scatter plot with smoothing fit of the pseudobulked gene expression of *Jun* against the binned amyloid-$\beta$ plaque density. Each dot is the pseudobulked expression per mouse, cell type, plaque density bin, and neighborhood status. The text at the top of the graph lists the number of cells from that cell type which were inside and outside of the *Jun* neighborhood.

**Extended Data Table 1 | Overview of the patients from whom the glioblastoma biopsies originated**

| Patient ID | Age | Gender | Tumor location | Condition | # Cells |
|---|---|---|---|---|---|
| PW030 | 65 | M | right parietal | vehicle (DMSO) | 17,384 |
| | | | | 0.2 µM panobinostat | 5,686 |
| PW032 | 61 | M | left frontal | vehicle (DMSO) | 5,003 |
| | | | | 0.2 µM panobinostat | 738 |
| PW034 | 68 | F | left parieto-occipital | vehicle (DMSO) | 12,375 |
| | | | | 0.2 µM panobinostat | 1,761 |
| PW036 | 56 | M | right temporal | vehicle (DMSO) | 10,292 |
| | | | | 0.2 µM panobinostat | 2,558 |
| PW040 | 69 | M | right temporal | vehicle (DMSO) | 8,901 |
| | | | | 0.2 µM panobinostat | 1,257 |

# Reporting Summary

## Statistics

For all statistical analyses, confirm that the following items are present in the figure legend, table legend, main text, or Methods section.

| n/a | Confirmed | |
|---|---|---|
| ☐ | ☒ | The exact sample size (*n*) for each experimental group/condition, given as a discrete number and unit of measurement |
| ☐ | ☒ | A statement on whether measurements were taken from distinct samples or whether the same sample was measured repeatedly |
| ☐ | ☒ | The statistical test(s) used AND whether they are one- or two-sided<br>*Only common tests should be described solely by name; describe more complex techniques in the Methods section.* |
| ☐ | ☒ | A description of all covariates tested |
| ☐ | ☒ | A description of any assumptions or corrections, such as tests of normality and adjustment for multiple comparisons |
| ☐ | ☒ | A full description of the statistical parameters including central tendency (e.g. means) or other basic estimates (e.g. regression coefficient) AND variation (e.g. standard deviation) or associated estimates of uncertainty (e.g. confidence intervals) |
| ☐ | ☒ | For null hypothesis testing, the test statistic (e.g. *F*, *t*, *r*) with confidence intervals, effect sizes, degrees of freedom and *P* value noted<br>*Give P values as exact values whenever suitable.* |
| ☒ | ☐ | For Bayesian analysis, information on the choice of priors and Markov chain Monte Carlo settings |
| ☒ | ☐ | For hierarchical and complex designs, identification of the appropriate level for tests and full reporting of outcomes |
| ☒ | ☐ | Estimates of effect sizes (e.g. Cohen's *d*, Pearson's *r*), indicating how they were calculated |

*Our web collection on statistics for biologists contains articles on many of the points above.*

## Software and code

Policy information about availability of computer code

| Data collection | No software was used for data collection |
|---|---|
| Data analysis | The complete code necessary to reproduce the results and analysis is available on https://github.com/const-ae/lemur-Paper (permanently stored with Zenodo (https://zenodo.org/doi/10.5281/zenodo.12726369)).<br><br>The software implementation of the proposed new method, LEMUR, is available open source, easy-to-install and well-documented at https://www.bioconductor.org/packages/lemur/ (doi: 10.18129/B9.bioc.lemur) as an R package and on https://pypi.org/project/pyLemur as a Python package.<br><br>We used LEMUR version 1.1.5, scVI version 1.1.2, CPA version 0.8.3, Harmony version 1.1.0, and miloDE version 0.0.9000 (hash: 8803302d). The version of all additional software dependencies can be found in the benchmark/renv.lock and benchmark/conda_env_info files. |

For manuscripts utilizing custom algorithms or software that are central to the research but not yet described in published literature, software must be made available to editors and reviewers. We strongly encourage code deposition in a community repository (e.g. GitHub). See the Nature Portfolio guidelines for submitting code & software for further information.

## Data

Policy information about availability of data

All manuscripts must include a data availability statement. This statement should provide the following information, where applicable:
- Accession codes, unique identifiers, or web links for publicly available datasets
- A description of any restrictions on data availability
- For clinical datasets or third party data, please ensure that the statement adheres to our policy

All datasets used in this manuscript are publicly available.

| Citation | Accession ID |
| --- | --- |
| Angelidis (2019) | GSE124872 |
| Aztekin (2019) | bioc::scRNAseq |
| Bunis (2021) | bioc::scRNAseq |
| Goldfarbmuren (2020) | GSE134174 |
| Hrvatin (2018) | GSE102827 |
| Jakel (2019) | GSE118257 |
| Sathyamurthy (2018) | GSE103892 |
| Kang (2018) | Zenodo 4473025 |
| Bhattacherjee (2019) | Zenodo 4473025 |
| Skinnider (2021) | Zenodo 4473025 |
| Cano (2020) | Zenodo 5048449 |
| Reyfman (2019) | Zenodo 5048449 |
| Pijuan (2019) | bioc::MouseGastrulationData |
| Zhao (2021) | GSE148842 |
| Saunders (2023) | GSE202639 |
| Cable (2022) | SCP1663 |

## Research involving human participants, their data, or biological material

Policy information about studies with human participants or human data. See also policy information about sex, gender (identity/presentation), and sexual orientation and race, ethnicity and racism.

| | |
| --- | --- |
| Reporting on sex and gender | We did not consider effects of sex or gender in the analysis of the glioblastoma data. There were too few individuals to draw any conclusions. Also, we have no reason to think that such effects, if they existed, would be relevant to the analysis we present. |
| Reporting on race, ethnicity, or other socially relevant groupings | N/A |
| Population characteristics | N/A |
| Recruitment | N/A |
| Ethics oversight | N/A |

Note that full information on the approval of the study protocol must also be provided in the manuscript.

# Field-specific reporting

Please select the one below that is the best fit for your research. If you are not sure, read the appropriate sections before making your selection.

☒ Life sciences          ☐ Behavioural & social sciences          ☐ Ecological, evolutionary & environmental sciences

For a reference copy of the document with all sections, see nature.com/documents/nr-reporting-summary-flat.pdf

# Life sciences study design

All studies must disclose on these points even when the disclosure is negative.

| | |
| --- | --- |
| Sample size | We chose publicy available datasets whose experimental designs, incl. sample size, were appropriate as a test case for the proposed method / underlying scientific question.. |
| Data exclusions | Across analyses, standard quality control filters were applied to set aside poor quality cells from the single-cell analysis. |
| Replication | We make all code available to make it easy to reproduce our analysis by third parties. We did not independently replicate the analysis ourselves. |

| Randomization | Not applicable. We present a complete combinatorial matrix of benchmarks of multiple computational methods each applied to multiple datasets and thus could observe each software in all conditions and did not need to randomize software to condition assignment. |
|---|---|
| Blinding | The analysts were not blinded while evaluating the benchmark. |

# Reporting for specific materials, systems and methods

We require information from authors about some types of materials, experimental systems and methods used in many studies. Here, indicate whether each material, system or method listed is relevant to your study. If you are not sure if a list item applies to your research, read the appropriate section before selecting a response.

## Materials & experimental systems

| n/a | Involved in the study |
|---|---|
| ☒ ☐ | Antibodies |
| ☒ ☐ | Eukaryotic cell lines |
| ☒ ☐ | Palaeontology and archaeology |
| ☒ ☐ | Animals and other organisms |
| ☒ ☐ | Clinical data |
| ☒ ☐ | Dual use research of concern |
| ☒ ☐ | Plants |

## Methods

| n/a | Involved in the study |
|---|---|
| ☒ ☐ | ChIP-seq |
| ☒ ☐ | Flow cytometry |
| ☒ ☐ | MRI-based neuroimaging |

## Plants

| Seed stocks | N/A |
|---|---|
| Novel plant genotypes | N/A |
| Authentication | N/A |

