## [Peer Review File · Nature Genetics]

Analysis of multi-condition single-cell data with latent embedding multivariate regression

Corresponding Author: Dr Wolfgang Huber

Version 0:

Decision Letter:

14th May 2024

Dear Wolfgang,

Your Article, "Analysis of multi-condition single-cell data with latent embedding multivariate regression" has now been seen by 3 referees. You will see from their comments below that while they find your work of interest, some important points are raised. We are interested in the possibility of publishing your study in Nature Genetics, but would like to consider your response to these concerns in the form of a revised manuscript before we make a final decision on publication.

In brief, the referees seem quite positive for the aims of LEMuR, but also make a range of requests for improvement.

Reviewer #1 is positive and says "there are no particular issues with the manuscript", suggesting they support publication as-is.

Referee #2 is also supportive, but makes two major requests: firstly, to compare to scVI in the head-to-head benchmarking; and secondly and more broadly/importantly, to improve the communication of the novelty and utility of LEMuR by e.g. improving the presentation for the journal's primarily non-technical audience, as well as strengthening the case studies. Reviewer #3 is again positive and provides a very long and thoughtful review, saying this is a "clever and elegant" approach. They also make some useful comments for improvement, intending to better-demonstrate the advance of your approach over the other tools out there.

In our reading, these are quite supportive reviews for a first round, and we think all of the specific requests made are reasonable. We would thus encourage you and your co-author to complete them all to the utmost. We would especially highlight Referee #2's request to improve the manuscript for our non-technical audience as especially important. We also agree with Reviewer #3 that their requests would strongly strengthen the work and we hope you will be motivated to do so!

To guide the scope of the revisions, the editors discuss the referee reports in detail within the team, including with the chief editor, with a view to identifying key priorities that should be addressed in revision and sometimes overruling referee requests that are deemed beyond the scope of the current study. We hope that you will find the prioritized set of referee points to be useful when revising your study. Please do not hesitate to get in touch if you would like to discuss these issues further.

We therefore invite you to revise your manuscript taking into account all reviewer and editor comments. Please highlight all changes in the manuscript text file. At this stage we will need you to upload a copy of the manuscript in MS Word .docx or similar editable format.

*2) If you have not done so already please begin to revise your manuscript so that it conforms to our Article format instructions, available

[here](http://www.nature.com/ng/authors/article_types/index.html).
Refer also to any guidelines provided in this letter.

*3) Include a revised version of any required Reporting Summary: <https://www.nature.com/documents/nr-reporting-summary.pdf>

Link Redacted

Sincerely,

Michael Fletcher, PhD
Senior Editor, Nature Genetics
ORCID: 0000-0003-1589-7087

Referee expertise: single-cell bioinformatics methods development.

Reviewers' Comments:

Reviewer #1:

Remarks to the Author:

In this manuscript, the authors present a new method to perform treatment vs control -style differential expression test of scRNA-seq data at the single cell level, enabling definition of populations of cells reacting to stimuli. This is in contrast to the typical workflow where populations of cells are first identified and then tested for treatment response.

The method allows simultaneous inference of low-dimensional representations of the cells with nuisance and treatment variation accounted for, and counterfactual effects for treatments of interest.

Compared to the standard workflow, this approach allows researchers to analyze their data with varying definitions of cell populations, which in practice often changes depending on context or question.

The authors demonstrate the use cases for their method in three different contexts, but also provide results from numerous benchmarks and calibration tests.

The primary alternative method compared against the proposed method is CPA. For the sub-task of integrating out nuisance factors, the proposed method is also compared to PCA as a baseline and Harmony as a competitive alternative. As a baseline for counterfactual prediction, the authors compare their method and CPA with 'identity prediction' which simply returns a null effect (unchanged expression). It would be more satisfying with some other baseline method to compare with, but for this case of single-cell level differential expression there does not seem to be alternatives in the literature. (Compare with the standard workflow of cell type definition followed by DE; in that case a typical baseline is to count the number of DE genes per cell type, but this would not apply in this case). Thus, it appears the fairly complex CPA method is the only available method to do a comparison with.

The manuscript is well-written and easy to follow. The mathematics (in particular in the methods section) might be on the abstract side for the target audience of the journal, but gives good references for further reading.

It is hard to think of improvements to the manuscript; benchmarking uses many proposed metrics, and are evaluated on a reasonable number of datasets and simulations. Figures reporting results are clear and comparisons with alternative methods are fair.

In sum, there are no particular issues with this manuscript.

Reviewer #2:

Remarks to the Author:

The paper presents a new model and corresponding software implementation to analyze multi-condition scRNA-seq data. The model includes terms for measurement error due to observed sources (technical covariates), expression variation due to cell type/state, and expression variation due to observed sources (perturbations).

The key innovations of the method are: (1) adapting the manifold regression method of Kim et al. 2014 to learn rotations of a common linear basis into condition-specific bases, and (2) identifying collections of single cells that have similar (counterfactual) DE effects rather than low distance in the latent space.

The proposed model shows promising results in benchmarking and case studies. The benchmarking results are strong; however, the case studies require more clarification for an audience of potential users of the method. Also, there are issues with the organization of the paper and explication of the method that need to be addressed.

Major points:

1. The organization of the paper could be improved to aid in reading and understanding the method for both expert and non-expert readers.

It would be helpful to present the benchmarks appear before the case studies, in order to first establish confidence in the method and then to apply it.

It may be helpful to discuss the geometric intuition shown in Fig S1A in more detail in the main text (section "Matrix factorization in the presence of known covariate information") in place of the more abstract discussion of eq. (2)-(4). I was better able to understand the meaning of notation such $R(X_{\{c\}})$ with in reference to affine transformation as opposed to the differential geometry discussion of eq. (10) in the methods.

Most of the technical material building up the full model could be moved from the methods section to a supplementary note, in order that expert readers can quickly find the actual objective functions and algorithm. These currently appear to be split between eqs. 13-16 and section "Implementation", respectively, and took me several readings to correctly locate and understand.

The related models described in the method could be moved to a supplementary note, and these relationships should be discussed in the Discussion section. A major oversight appears to be the relation to scVI and derivative methods.

A link to the code to (re)produce the analysis results in the paper should be included in the main text, and ideally archived in Zenodo.

2. There are some omissions in the description of the method.

l. 82 "In addition, it expects two tables of metadata for the cells: the vector F of length C which for each cell specifies the sample (independent experimental unit, e.g., tissue biopsy or organoid) it originates from": I cannot find F in any of the equations. Where is F used in the model?

Section "Non-distance preserving extension": It is not clear from the text

at what point in the algorithm described in section "Implementation" this is fit.

In eq. (13), how is λ set?

3. A major oversight in the benchmarking appears to be comparison against scVI, which can also harmonize the data across conditions (as defined in this paper) by embedding into a common space and can also produce counterfactual true gene expression levels given the observed data and a target condition. I think this comparison is critical to support the claim that linear methods work equally well or better than deep learning-based methods.

4. The main figures are too dense to understand properly, and the discussion in the main text needs additional elaboration to explain to a non-expert reader what the main biological takeaways are.

In Fig 2, additional clarification in the main text is needed to explain the interpretation and importance of panel (D). Panel 2(F) appears to be the main result but is not so clearly delineated as such in the main text.

Fig. 3 appears to contain results for two separate case studies, and should be split into two figures. The selection of the genes to highlight in Fig 3C needs more motivation and explanation in the main text; are these genes already known show temporal patterns? Does the method reveal temporal patterns that weren't already known?

Fig. 4B should include the original data points in addition to the density contours. In the axis labels, I do not follow what is meant by "relative".

I don't understand what is plotted in Fig. 4H. Are they precision and recall at some fixed FDR? How were they computed?

I am unable to reconcile the results in Fig 4J, which the paper interprets as showing LEMUR has higher power than miloDE across all data sets, with the results in Fig 4K, which do not obviously show that LEMUR discovers more of the simulated DE genes than miloDE. Have I misunderstood what is being plotted?

Minor points

1. l. 49 "For differential expression analysis, state-of-the-art methods take an integrated embedding, discretize it into (potentially overlapping) "clusters", and run a pseudobulked differential expression test for each cluster separately"

I do not think this accurately describes scVI and derivative methods, which marginalize over the posterior distribution of log fold changes between pairs of cells in two groups.

2. l. 94 "We term a unique combination of covariate values a condition"

It would be helpful to make explicit that covariates include both sources of unwanted variation (e.g., age, sex) and sources of variation whose effect is to be assessed (e.g., treatment).

Related, the proposed model does not appear to include a component for unwanted variation due to unobserved/unrecorded covariates. In bulk RNA-seq, such a component would typically be learned by SVA/PEER. Would the presence of such covariates bias the method? Is it straightforward to extend the method to include an SVA-like component?

3. l. 302: "we found that panobinostat caused cell population-specific expression changes"

Is this the same cell population for each gene? The following paragraph suggests not.

It seems in this case study that one would like to identify coherent subpopulations of cells, and the coherent sets of genes indicating

differential responses to panobinostat, in order to understand the biology of differential response.

Is this possible from the LEMUR results? If not, how should a user approach a method that can indicate DE in a different set of focal cells for each DE gene (in the worst case)?

4. l. 949: "Principal component analysis (PCA) is a special case, where in addition the columns of R are obtained from an eigendecomposition and ordered by eigenvalue."

This is confusing. I agree that PCA is a special case, but one where Z is required to be orthogonal in addition to R orthonormal (c.f. Srebro and Jaakola 2003, Engelhardt and Stephens 2010).

5. In eq. (36), what is sf_c ? Is it a size factor? How was it simulated?

6. Fig S5 (B): it would be helpful to make the y-axis limits from 0 to 1 for each row.

7. Fig S5 (C): "ration" is a typo

Reviewer #3:

Remarks to the Author:

The authors develop, document, and demonstrate the use and utility of their LEMuR (Latent Embedding Multivariate Regression) method for integration of multi-condition, multi-dataset, or multi-modal single-cell data. There is no particular reason the method could not be used for bulk samples with these properties, but single-cell analysis is fashionable at present, so that's what's done. The documentation for the package implementing LEMuR has radically improved in the past week alone, and the authors clearly indicate the intended use case (predicting the impact of a condition on gene expression using latent space regression with held-out validation of prediction error in a ground-truth exposed population, without depending upon discrete clusters). This is a worthwhile approach due to its relative ease of interpretation; presumably it is less frequently considered due to the cargo cult tendencies of users to cluster scRNAseq data.

The authors begin the manuscript with a warning about premature discretization and show that manifold alignment combined with pseudobulk randomized and balanced comparisons can address the issues of both integration and discretization, further showing that parceling out batch or technical effects leads to integration performance on a par with state-of-the-art methods such as Harmony (without the mixture-of-experts overhead).

A) Key results: traditional latent space methods derived from the singular value decomposition (SVD) or rotations thereof (i.e., PCA and factor analysis) encounter difficulties disentangling biologically interesting variation from technical or batch variation. LEMuR employs a parametric model on log-transformed or otherwise variance-stabilized abundance data to disambiguate and partition the variance explicitly, while employing a manifold alignment to address nonlinear technical artifacts. This is a clever and elegant approach to two very nasty problems.

B) originality and significance: to my knowledge, previous applications of Grassmann manifolds for high-dimensional overdetermined datasets in latent space is new. This is the crux of the technique and represents a Gordian knot type of approach, slicing through a great many complexities in one fell swoop.

C) the authors present the results on multiple datasets, using ribosomal (translational upregulation in myeloid cells panobinostat-treated glioma samples) differential expression and cross-prediction as one example, zebrafish development as another, mouse gastruloids as another, and IFNB-treated SLE patients as another) to construct a rather dense but highly informative figure 4, benchmarking the perturbation predictions against simulation (of particular interest).

Here I must stop for a moment and consider the role of LEMUR in a space crowded by deep generative models, and ponder the efficiency of predicting (e.g.) CRISPR-induced expression perturbation with LEMUR as opposed to (say) scGPT or GeneFormer. The latter two are fashionable but excruciatingly inefficient (scGPT in particular trains 96 transformers in parallel and requires hyperparameter fine-tuning for condition-specific predictions). A study from Microsoft Research (<https://www.biorxiv.org/content/10.1101/2023.10.16.561085v2.full>) suggests that relatively simple models (scVI, in particular) perform favorably to Transformer-based foundation models, and this begs the question of "how far can we simplify before losing appreciable performance?" I believe that it would be of significant interest to Nature Genetics readers to know whether simpler parametric models such as LEMUR may be able to predict responses to perturbations such as CRISPR screens in addition to (e.g.) panobinostat treatment. To me, this represents the key Additional Experiment (tm) that would distinguish the manuscript from a great many Methods Papers that do not address a key and timely topic.

D) stats: the authors are statisticians and the derivations appear sound. many diverse benchmarking datasets are used.

E) Conclusions: see above. I want to know whether LEMUR can save me \$3000 on a CRISPR screen!

F) See above.

G) Copious references to previous work

H) Figures can be improved, mostly I suspect readers would dearly love to see if LEMUR can predict screen results (!)

This is a significant manuscript regarding a significant methodological advance from significant authors. I would dearly love to see it subjected to significant stress-testing and determine how an unfashionable (but fast and interpretable) technique compares to fashionable (but extremely inefficient) foundation model production methods currently on parade. It will be of broad and significant interest for investigators to know whether certain screens or experimental designs can be piloted _in silico_ and with minor technical difficulty when compared to (e.g.) training a transformer on the HCA.

Version 1:

Decision Letter:

Our ref: NG-A65020R

29th Aug 2024

Dear Wolfgang,

Thank you for submitting your revised manuscript "Analysis of multi-condition single-cell data with latent embedding multivariate regression" (NG-A65020R). It has now been seen by the original referees and their comments are below. The reviewers find that the paper has improved in revision, and therefore we'll be happy in principle to publish it in Nature Genetics, pending minor revisions to satisfy the referees' final requests and to comply with our editorial and formatting guidelines.

Sincerely,

Michael Fletcher, PhD
Senior Editor, Nature Genetics
ORCID: 0000-0003-1589-7087

Reviewer #1 (Remarks to the Author):

In this revision, the authors have greatly improved the readability of the manuscript. They have also included the scVI method as an additional alternative approach which is compared against.

There are no outstanding issues.

Reviewer #2 (Remarks to the Author):

I have no remaining issues with the manuscript.

Reviewer #3 (Remarks to the Author):

A. Key results: LEMUR complements methods such as SVA or PEER and outperforms a number of competing generative approaches, while retaining an invertible transformation that allows neighborhood-wise differential expression analysis (in addition to the possibly more straightforward approach of factor-wise differential expression analysis) without prespecified clustering or grouping. The method determines neighborhoods of interest for comparisons as a byproduct of aligning manifolds and decomposing latent variance into experimental, explained non-experimental, and unwanted. The ability to decode the aligned results linearly significantly improves interpretability. The authors further show that these improvements extend to spatial and multi-view analyses.

- B. This approach is novel and significantly less complex than many competing generative models. The authors also demonstrate that it outperforms most competing models in most situations via a series of careful comparisons that may form the basis for a separate manuscript.
- C. Data and methods choices are excellent.
- D. The authors are statisticians and the math is exceptionally detailed in the supplement.
- E. I am satisfied that the authors' initial claims are general, and that aligned latent linear models can compete with or even outperform far more complicated generative models in practice.
- F. None noted.
- G. The authors appear to have added citations (and methods, e.g. scVI) where appropriate.
- H. The work is quite clear. It could be made more impactful by stating the crux of the claims but this is the authors' paper, not mine.

Point-by-point response to reviewers

Analysis of multi-condition single-cell data with latent embedding multivariate regression

Constantin Ahlmann-Eltze and Wolfgang Huber

2024-07-29

Reviewer 1

In this manuscript, the authors present a new method to perform treatment vs control -style differential expression test of scRNA-seq data at the single cell level, enabling definition of populations of cells reacting to stimuli. This is in contrast to the typical workflow where populations of cells are first identified and then tested for treatment response.

The method allows simultaneous inference of low-dimensional representations of the cells with nuisance and treatment variation accounted for, and counterfactual effects for treatments of interest.

Compared to the standard workflow, this approach allows researchers to analyze their data with varying definitions of cell populations, which in practice often changes depending on context or question.

The authors demonstrate the use cases for their method in three different contexts, but also provide results from numerous benchmarks and calibration tests.

The primary alternative method compared against the proposed method is CPA. For the sub-task of integrating out nuisance factors, the proposed method is also compared to PCA as a baseline and Harmony as a competitive alternative. As a baseline for counterfactual prediction, the authors compare their method and CPA with ‘identity prediction’ which simply returns a null effect (unchanged expression). It would be more satisfying with some other baseline method to compare with, but for this case of single-cell level differential expression there does not seem to be alternatives in the literature. (Compare with the standard workflow of cell type definition followed by DE; in that case a typical baseline is to count the number of DE genes per cell type, but this would not apply in this case). Thus, it appears the fairly complex CPA method is the only available method to do a comparison with.

The manuscript is well-written and easy to follow. The mathematics (in particular in the methods section) might be on the abstract side for the target audience of the journal, but gives good references for further reading.

It is hard to think of improvements to the manuscript; benchmarking uses many pro-

posed metrics, and are evaluated on a reasonable number of datasets and simulations. Figures reporting results are clear and comparisons with alternative methods are fair. In sum, there are no particular issues with this manuscript.

We thank the reviewer for their time and effort they spent on our manuscript, and for their positive feedback.

Regarding comparison to other methods, upon further reflection and a comment by Reviewer 2, we have now also added a comparison to *scVI* (see Figs. 3, S1, S3).

Reviewer 2

Remarks to the Author:

The paper presents a new model and corresponding software implementation to analyze multi-condition scRNA-seq data. The model includes terms for measurement error due to observed sources (technical covariates), expression variation due to cell type/state, and expression variation due to observed sources (perturbations).

The key innovations of the method are: (1) adapting the manifold regression method of Kim et al. 2014 to learn rotations of a common linear basis into condition-specific bases, and (2) identifying collections of single cells that have similar (counterfactual) DE effects rather than low distance in the latent space.

The proposed model shows promising results in benchmarking and case studies. The benchmarking results are strong; however, the case studies require more clarification for an audience of potential users of the method. Also, there are issues with the organization of the paper and explication of the method that need to be addressed.

We thank the reviewer for their time and effort they spent on our manuscript. We also thank them for their positive feedback and for their constructive comments on improving the presentation of our work.

Major points:

1. The organization of the paper could be improved to aid in reading and understanding the method for both expert and non-expert readers.

It would be helpful to present the benchmarks appear before the case studies, in order to first establish confidence in the method and then to apply it.

We have adopted this suggestion and reorganized the presentation of the results accordingly.

It may be helpful to discuss the geometric intuition shown in Fig S1A in more detail in the main text (section “Matrix factorization in the presence of known covariate information”) in place of the more abstract discussion of eq. (2)-(4). I was better able to understand the meaning of notation such $R(X_{\{c:\}})$ with in reference to affine transformation as opposed to the differential geometry discussion of eq. (10) in the methods.

Most of the technical material building up the full model could be moved from the methods section to a supplementary note, in order that expert readers can quickly find the actual objective functions and algorithm. These currently appear to be split between eqs. 13-16 and section “Implementation”, respectively, and took me several readings to correctly locate and understand.

We have taken up these suggestions and reorganized the paper accordingly. In particular:

- The *Results* section in the main text now works without using display style equations, and with minimal mathematical notation. We have promoted and revised Supplementary Figure S1A from the previous version of the manuscript into the new Figure 2 in the main text, following the reviewer’s suggestion.
- The formal presentation of the core method using display style equations is now consolidated into a single place, in the *Methods* section. We consider this the main piece of scholarship in this paper, upon which the application examples and the software are based. It is also the main point of reference for potential future work that aims to build upon or modify our method.
- The section *Implementation*, which was previously in the main text, has been moved into a Supplementary Note, and also into the documentation of the software package.

We hope that with these comprehensive changes, we have taken into account the reviewers’ suggestions and improved the readability and navigatability of the paper.

The related models described in the method could be moved to a supplementary note, and these relationships should be discussed in the Discussion section. A major oversight appears to be the relation to scVI and derivative methods.

We thank the reviewer for pointing out that we had missed to note the relevance of *scVI* to our work. We have now included *scVI* in the performance validation section (now Figure 3, and Supplementary Figures S1 and S3). It turns out that it performs only slightly better than the no-effort baseline, i.e., predicting no change (Fig. 3D, *scVI* vs. Identity).

We have also followed the reviewer’s suggestion to move the discussion of related models to a supplementary note, which we refer to in the Discussion section.

A link to the code to (re)produce the analysis results in the paper should be included in the main text, and ideally archived in Zenodo.

We archived the code using Zenodo (doi.org/10.5281/zenodo.12726370) and included a link to the archive in the *Availability* section before the *References*.

2. There are some omissions in the description of the method.

1. 82 “In addition, it expects two tables of metadata for the cells: the vector F of length C which for each cell specifies the sample (independent experimental unit, e.g., tissue biopsy or organoid) it originates from”: I cannot find F in any of the equations. Where is F used in the model?

We have removed the reference to the vector F . The assignment of cells to samples is only relevant for the pseudobulking, which we describe in Eqn. (22).

Section “Non-distance preserving extension”: It is not clear from the text at what point in the algorithm described in section “Implementation” this is fit.

The non-distance preserving extension is an optional step that can be run in addition to fitting the distance preserving model.

In eq. (13), how is λ set?

We have empirically noted that a tuning factor of $\lambda = 0.01$ worked well across a wide range of different datasets. We allow the user to change λ , but in our experience is not critical for the performance. Future research might find ways to fully automate an appropriate choice of λ . We updated the text in the *Non-distance preserving extension* subsection to better explain these two points.

3. A major oversight in the benchmarking appears to be comparison against scVI, which can also harmonize the data across conditions (as defined in this paper) by embedding into a common space and can also produce counterfactual true gene expression levels given the observed data and a target condition. I think this comparison is critical to support the claim that linear methods work equally well or better than deep learning-based methods.

Thank you for pointing this out. See the response to Point 1.

4. The main figures are too dense to understand properly, and the discussion in the main text needs additional elaboration to explain to a non-expert reader what the main biological takeaways are.

We thank the reviewer for their feedback. We have gone through the figures and simplified them, to better highlight and concentrate on the most important biological takeaways:

- We simplified the bottom two rows of the performance assessment figure, removing three subpanels,
- we moved three subpanels from the glioblastoma figure to the Supplementary Figures,
- we split the Zebrafish and Alzheimer figures into two separate figures, which makes each individually easier to understand.

In Fig 2, additional clarification in the main text is needed to explain the interpretation and importance of panel (D). Panel 2(F) appears to be the main result but is not so clearly delineated as such in the main text.

We thank the reviewer for their feedback. We have moved panels C-E to a Supplementary Figure. This simplifies the figures and emphasizes that, indeed, panel (F) is the main result of the glioblastoma analysis.

Fig. 3 appears to contain results for two separate case studies, and should be split into two figures. The selection of the genes to highlight in Fig 3C needs more motivation and explanation in the main text; are these genes already known show temporal patterns? Does the method reveal temporal patterns that weren't already known?

We have split the previous Fig. 3 into two separate figures for Zebrafish and Alzheimer’s, as suggested by the reviewer.

We have amended the main text to more clearly explain how we selected the four exemplary genes. The selection was data-driven, but these genes all turned out to be known to play a role in the relevant cell types and in development. To the best of our knowledge, the specific patterns that we describe here, divergences in the temporal developmental profile within the same cell type but between different cell ‘states’ have not been described, but are plausible, e.g., due to spatial polarities such as anterior-posterior, dorsal-ventral. Because the embryos analyzed by Saunders were dissociated, there is no direct way to verify this from their data. More detailed follow-up experiments might be interesting contributions to developmental biology and/or the functional study of these genes, but would in our view add little to the presentation of the LEMUR method. Here, the main point is that it is for the first time possible to systematically search for and discover such patterns.

Fig. 4B should include the original data points in addition to the density contours. In the axis labels, I do not follow what is meant by “relative”.

We now plot the results for each of the thirteen datasets as points (with reduced opacity) in addition to the density of the mean estimate. We also updated the description in the Figure legend to better explain what is meant by relative k -NN and relative ARI. In the process, we realized that our original way to calculate the ARI favored the PCA-based approaches (PCA, LEMUR, and Harmony). We have now fixed this as well and updated the description accordingly.

I don’t understand what is plotted in Fig. 4H. Are they precision and recall at some fixed FDR? How were they computed?

We thank the reviewer for their feedback. We have decided to remove panel H as it added little new.

I am unable to reconcile the results in Fig 4J, which the paper interprets as showing LEMUR has higher power than miloDE across all data sets, with the results in Fig 4K, which do not obviously show that LEMUR discovers more of the simulated DE genes than miloDE. Have I misunderstood what is being plotted?

Unlike LEMUR, which performs a single differential expression test per gene, miloDE performs many tests for each gene, namely one for each neighborhood (Typically, there are 10 to 50 neighborhoods depending on the complexity of the data). In the old Fig 4I, the performance was measured using the true positive rate, which meant that the denominator of the TPR differed between LEMUR and miloDE. In contrast, in Fig 4J, we showed the number of genes for which miloDE considered at least one neighborhood differentially expressed. This approach to aggregate the results across neighborhoods into a single per-gene result favors miloDE in terms of recall, however, it also leads to catastrophic loss of specificity or type-I error control. We realized that altogether this was too confusing and incoherent, and we decided to remove the stratification by neighborhood size (i.e., old panel J). We have

further simplified the presentation of the FDR control presentation and the new Fig. 3G now only shows the type I error control and power for a nominal FDR of 10%. The more detailed presentation is still available in Suppl. Fig. S4.

Minor points

1. l. 49 “For differential expression analysis, state-of-the-art methods take an integrated embedding, discretize it into (potentially overlapping) “clusters”, and run a pseudobulked differential expression test for each cluster separately”

I do not think this accurately describes scVI and derivative methods, which marginalize over the posterior distribution of log fold changes between pairs of cells in two groups.

To the best of understanding, scVI is focused on finding gene expression differences between cell types, but not between experimental conditions (scVI-tools Development Team 2024; Boyeau et al. 2019). Furthermore, to our knowledge, none of the tools from the scVI family account for the replication structure of the data (e.g., which cells come from the same, or different, replicate samples for a condition), although this is crucial for reliable uncertainty quantification in differential expression analysis (Crowell et al. (2020); 100 patient samples with 1000 cells each enable different conclusions than 1 sample with 100000 cells.) In addition, scVI and its derivative methods like lvm-DE still need for the comparison between cell population A and B, a discrete assignment of cells to the respective subpopulations (Boyeau et al. 2023).

2. l. 94 “We term a unique combination of covariate values a condition”

It would be helpful to make explicit that covariates include both sources of unwanted variation (e.g., age, sex) and sources of variation whose effect is to be assessed (e.g., treatment).

Related, the proposed model does not appear to include a component for unwanted variation due to unobserved/unrecorded covariates. In bulk RNA-seq, such a component would typically be learned by SVA/PEER. Would the presence of such covariates bias the method? Is it straightforward to extend the method to include an SVA-like component?

We want to thank the reviewer for this suggestion. We included a sentence in that paragraph that explains that the design matrix can reflect known “nuisance” covariates that we simply want to adjust for, and “interesting” covariates whose effects we want to test for and characterize.

The underlying idea of SVA and PEER is to identify surrogate variables by finding shared latent variation across samples and assume that these represent unwanted batch effects. In contrast, LEMUR assumes that the latent variation represents different cell types and states.

At this point, nothing speaks against using SVA and PEER subsequent to LEMUR, on pseudobulk summaries of the LEMUR output, infer surrogate variables, and then use these as explicit covariates in an iterated application of LEMUR.

A more ambitious approach would be to combine these ideas into an integrated model, and add latent variation at the sample level to the LEMUR model, using SVA- or PEER-style surrogate variables. A substantially more complicated inference procedure would be required to account for such a multi-level (sample and cells) latent structure of the data. We think that it is too early to add such further complexity to LEMUR, and would first like to see its usability for a broad range of users in its current form.

In a properly designed experiment, unwanted variation leads to extra “noise” and thus a loss of power, but not to bias and increased false discoveries. Thus, the main benefit of modeling unwanted variation with approaches such as SVA and PEER is potentially increased power to detect differential expression. Including such a component into LEMUR is an interesting direction for future research, but beyond the scope of the current work.

3. 1. 302: “we found that panobinostat caused cell population-specific expression changes”

Is this the same cell population for each gene? The following paragraph suggests not. It seems in this case study that one would like to identify coherent subpopulations of cells, and the coherent sets of genes indicating differential responses to panobinostat, in order to understand the biology of differential response.

Is this possible from the LEMUR results? If not, how should a user approach a method that can indicate DE in a different set of focal cells for each DE gene (in the worst case)?

The reviewer correctly points out that the neighborhoods are currently found separately for each gene, and their consolidation across genes is up to the user. We agree that this is a useful postprocessing step, which can, for example, be achieved by clustering the neighborhoods (e.g., measure set similarity by Jaccard index and call a distance-based clustering method such as hierarchical clustering). We could also imagine more sophisticated approaches, where neighborhood relations such as equality and subset are already integrated into LEMUR’s inference and “information sharing across genes” takes place. This is a promising and substantial future research program for us and others, which we hope the publication of the current work will enable.

4. 1. 949: “Principal component analysis (PCA) is a special case, where in addition the columns of R are obtained from an eigendecomposition and ordered by eigenvalue.”

This is confusing. I agree that PCA is a special case, but one where Z is required to be orthogonal in addition to R orthonormal (c.f. Srebro and Jaakola 2003, Engelhardt and Stephens 2010).

We updated the text to mention both definitions.

5. In eq. (36), what is sf_c ? Is it a size factor? How was it simulated?

Yes, sf_c are size factors, which we calculated using the expression of the non-simulated genes for that cell (i.e., we did not simulate them). We updated the text around what is now Eq. (24) and mention how the size factors were calculated.

6. Fig S5 (B): it would be helpful to make the y -axis limits from 0 to 1 for each row.

We changed this figure to be overall more legible. In particular, we have adjusted the y -axis to go from 0 to 1 (see Suppl. Fig. S1 in current version).

7. Fig S5 (C): “ration” is a typo

Fixed.

Reviewer 3

The authors develop, document, and demonstrate the use and utility of their LEMuR (Latent Embedding Multivariate Regression) method for integration of multi-condition, multi-dataset, or multi-modal single-cell data. There is no particular reason the method could not be used for bulk samples with these properties, but single-cell analysis is fashionable at present, so that's what's done. The documentation for the package implementing LEMuR has radically improved in the past week alone, and the authors clearly indicate the intended use case (predicting the impact of a condition on gene expression using latent space regression with held-out validation of prediction error in a ground-truth exposed population, without depending upon discrete clusters). This is a worthwhile approach due to its relative ease of interpretation; presumably it is less frequently considered due to the cargo cult tendencies of users to cluster scRNAseq data.

The authors begin the manuscript with a warning about premature discretization and show that manifold alignment combined with pseudobulk randomized and balanced comparisons can address the issues of both integration and discretization, further showing that parceling out batch or technical effects leads to integration performance on a par with state-of-the-art methods such as Harmony (without the mixture-of-experts overhead).

A) Key results: traditional latent space methods derived from the singular value decomposition (SVD) or rotations thereof (i.e., PCA and factor analysis) encounter difficulties disentangling biologically interesting variation from technical or batch variation. LEMuR employs a parametric model on log-transformed or otherwise variance-stabilized abundance data to disambiguate and partition the variance explicitly, while employing a manifold alignment to address nonlinear technical artifacts. This is a clever and elegant approach to two very nasty problems.

B) originality and significance: to my knowledge, previous applications of Grassmann manifolds for high-dimensional overdetermined datasets in latent space is new. This is the crux of the technique and represents a Gordian knot type of approach, slicing through a great many complexities in one fell swoop.

C) the authors present the results on multiple datasets, using ribosomal (translational upregulation in myeloid cells panobinostat-treated glioma samples) differential expression and cross-prediction as one example, zebrafish development as another, mouse gastruloids as another, and IFNB-treated SLE patients as another) to construct a rather dense but highly informative figure 4, benchmarking the perturbation predictions against simulation (of particular interest).

Here I must stop for a moment and consider the role of LEMUR in a space crowded by deep generative models, and ponder the efficiency of predicting (e.g.) CRISPR-induced expression perturbation with LEMUR as opposed to (say) scGPT or GeneFormer. The latter two are fashionable but excruciatingly inefficient (scGPT in particular trains 96 transformers in parallel and requires hyperparameter fine-tuning for condition-specific predictions). A study from Microsoft Research (<https://www.biorxiv.org/content/10.1101/2023.10.16.561085v2.full>) suggests that

relatively simple models (scVI, in particular) perform favorably to Transformer-based foundation models, and this begs the question of “how far can we simplify before losing appreciable performance?” I believe that it would be of significant interest to Nature Genetics readers to know whether simpler parametric models such as LEMUR may be able to predict responses to perturbations such as CRISPR screens in addition to (e.g.) panobinostat treatment. To me, this represents the key Additional Experiment (tm) that would distinguish the manuscript from a great many Methods Papers that do not address a key and timely topic.

D) stats: the authors are statisticians and the derivations appear sound. many diverse benchmarking datasets are used.

E) Conclusions: see above. I want to know whether LEMUR can save me \$3000 on a CRISPR screen!

F) See above.

G) Copious references to previous work

H) Figures can be improved, mostly I suspect readers would dearly love to see if LEMUR can predict screen results (!)

This is a significant manuscript regarding a significant methodological advance from significant authors. I would dearly love to see it subjected to significant stress-testing and determine how an unfashionable (but fast and interpretable) technique compares to fashionable (but extremely inefficient) foundation model production methods currently on parade. It will be of broad and significant interest for investigators to know whether certain screens or experimental designs can be piloted *in silico* and with minor technical difficulty when compared to (e.g.) training a transformer on the HCA.

We thank the reviewer for their encouraging feedback, and for these thoughtful and inspiring suggestions. We share their preference for what has been called “Occam’s razor”; in this case, that methods should not be more complicated for the task at hand than necessary. We also become skeptical when tools are presented for the sake of their complexity and complicatedness, rather than utility or usability. Before proceeding to our specific reply, we would however like to note a fundamental question of scope:

- Foundations models such as those trained on the HCA would try to capture as much as possible of all existing data. They try to make everything that is knowable about biology from the HCA available in a computational form. When faced with new data, they would generate responses that somehow amalgamate the previously known with the new. Perhaps their main utility is in just responding to biological queries without any new data.
- LEMUR looks only at the data at hand. It does not have a background knowledge of other data. The conclusions it arrives at are purely driven by the new data.

Both activities are scientifically valuable. The former approach is perhaps somewhat comparable to a clever database lookup, the latter to the generation of new knowledge from new data. Good scientific discovery research will combine both, perhaps at different stages of a project.

That said, we took up the reviewer’s and the editor’s suggestion and set up a comparison of LEMUR with other tools for predicting the effect of perturbation experiments. An important caveat about available perturbation screen data is that these were done on homogeneous populations of cells from a cell line. In this case, there is no outspoken need for LEMUR’s ability to model latent variation (such as from different cell types) and to test how it affects the response to the perturbation. We nonetheless compared the performance of ordinary linear models, LEMUR, and the deep learning-based scGPT (Cui et al. 2024) and GEARS (Roohani, Huang, and Leskovec 2024) in several benchmarks. As detailed in the following, simple linear models outperformed the deep learning approaches. We believe that these results are very interesting, and we would like to publish them. However, we believe that this analysis is too removed from the core message of the current manuscript and would be better presented as a separate publication, in terms of the amount of detail needed to describe it, but also in terms of audience and its own message.

We did not include Geneformer (Theodoris et al. 2023) in our comparison, as it is not a generative model. Geneformer can predict the changes of the cell embedding after an *in silico* perturbation, but cannot predict the corresponding gene expression changes.

Predicting the effect of double perturbations from single perturbations

On a dataset of gene expression profiles from K562 cells after single- and double CRISPRi perturbations (Norman et al. 2019), both scGPT and GEARS performed considerably worse than a simple additive model to predict the effect of the double perturbations from the single perturbations (Figure 1 A, first panel from left). As LEMUR is built around a design matrix and linear modeling ideas, its performance was on par with the additive model. To confirm that our results are not due to a method operation error on our side, we approached the authors of GEARS, who confirmed that they are aware of this type of finding and pointed us to a paper by Gaudalet et al. (2024) with similar findings.

We found that GEARS and scGPT sometimes correctly identified non-additive perturbation effects, but more often falsely predicted them (Figure 1 B).

Predicting the effect of unseen single perturbations

Predicting the effect of a new perturbation necessarily requires some sort of information on how it relates to already seen perturbations. GEARS and scGPT both focus on predicting genetic perturbations. GEARS uses gene ontology annotations, and scGPT is pretrained on the human cell atlas. As an alternative, we devised a simplistic linear model that “learns” gene function by mapping each gene to a low-dimensional linear space based on coexpression (expression correlation) in the training data and then predicts perturbation phenotype using linear inter- or extrapolation (details in Methods). This model performed as good as GEARS on the Adamson data and outperformed GEARS and scGPT on the Replogle data (see Datasets section for more details). Furthermore, a simple linear model pretrained on reference data with overlapping perturbations, outperformed all other methods for predicting the effect of the same perturbations in a new setting (Figure 1 A, panel 2-4).

Figure 1: Comparison of perturbation effect predictions. (A) Beeswarm plots of the Pearson correlation between predicted and observed change compared to the unperturbed condition, for four methods and four datasets. Each point is the correlation coefficient of gene expression profiles for one perturbation, the horizontal red line is their mean. (B) Scatter plot of true interaction effect, i.e., the difference between the observed value under double perturbation and the sum of single perturbation effects (x -axis), versus predicted interaction, i.e., the difference between model prediction and the sum of single perturbation effects (y -axis), for GEARs and scGPT. The dashed diagonal line indicates optimal predictions. The numbers count the number of points in each of the five segments. (C) First two dimensions of the PCA embedding of the perturbation in the RPE1 data (left) and the aligned perturbations from K562 (right).

Figure 1 C illustrates the pretraining process. The perturbation responses from the RPE1 data were used for training, and perturbations from the K562 data were aligned with them. The linear and the pretrained linear methods have two tuning parameters: the number of dimensions for the embedding and the ridge penalty. The optimal choice depends on the dataset, but we find empirically that both methods are robust across a wide range of different parameters (Figure 2).

Conclusion

We show that, as the reviewer hypothesized, recently presented deep-learning approaches do not yet provide a performance benefit for predicting the effect of perturbations over quite

simple linear methods.

LEMUR performs as good as these simple linear methods, but does not provide additional performance benefits, mainly because there is no need for its capability to model latent “cell type” or “cell state” heterogeneity in cell line experiments. We expect that once this need becomes relevant, i.e., when perturbation responses from organoids or primary patient samples are recorded using sc-RNA-seq or analogous measurement assays, LEMUR might have a unique edge. To date, we have however not yet identified suitable benchmark data that were systematically produced on a sufficient number of perturbations.

We show that simple linear methods are already able to accurately predict the gene expression in a new setting if a good reference dataset is available. This has ramifications for experimental design: measuring many perturbations in a few cell lines can enable prediction of perturbation effects in cell lines with only a small number of perturbations.

References

- Adamson, Britt, Thomas M Norman, Marco Jost, Min Y Cho, James K Nuñez, Yuwen Chen, Jacqueline E Villalta, et al. 2016. “A Multiplexed Single-Cell CRISPR Screening Platform Enables Systematic Dissection of the Unfolded Protein Response.” *Cell* 167 (7): 1867–82.
- Boyeau, Pierre, Romain Lopez, Jeffrey Regier, Adam Gayoso, Michael I Jordan, and Nir Yosef. 2019. “Deep Generative Models for Detecting Differential Expression in Single Cells.” *bioRxiv*, 794289.
- Boyeau, Pierre, Jeffrey Regier, Adam Gayoso, Michael I Jordan, Romain Lopez, and Nir Yosef. 2023. “An Empirical Bayes Method for Differential Expression Analysis of Single Cells with Deep Generative Models.” *Proceedings of the National Academy of Sciences* 120 (21): e2209124120.
- Crowell, Helena L, Charlotte Soneson, Pierre-Luc Germain, Daniela Calini, Ludovic Collin, Catarina Raposo, Dheeraj Malhotra, and Mark D Robinson. 2020. “Muscat Detects Subpopulation-Specific State Transitions from Multi-Sample Multi-Condition Single-Cell Transcriptomics Data.” *Nature Communications* 11 (1): 6077. <https://doi.org/10.1038/s41467-020-19894-4>.
- Cui, Haotian, Chloe Wang, Hassaan Maan, Kuan Pang, Fengning Luo, Nan Duan, and Bo Wang. 2024. “scGPT: Toward Building a Foundation Model for Single-Cell Multi-Omics Using Generative AI.” *Nature Methods*, 1–11.
- Gaudelet, Thomas, Alice Del Vecchio, Eli M Carrami, Juliana Cudini, Chantriolnt-Andreas Kapourani, Caroline Uhler, and Lindsay Edwards. 2024. “Season Combinatorial Intervention Predictions with Salt & Peper.” *arXiv Preprint arXiv:2404.16907*.
- Norman, Thomas M, Max A Horlbeck, Joseph M Replogle, Alex Y Ge, Albert Xu, Marco Jost, Luke A Gilbert, and Jonathan S Weissman. 2019. “Exploring Genetic Interaction Manifolds Constructed from Rich Single-Cell Phenotypes.” *Science* 365 (6455): 786–93.
- Replogle, Joseph M, Reuben A Saunders, Angela N Pogson, Jeffrey A Hussmann, Alexander Lenaill, Alina Guna, Lauren Mascibroda, et al. 2022. “Mapping Information-Rich Genotype-Phenotype Landscapes with Genome-Scale Perturb-Seq.” *Cell* 185 (14): 2559–75.

- Roohani, Yusuf, Kexin Huang, and Jure Leskovec. 2024. “Predicting Transcriptional Outcomes of Novel Multigene Perturbations with GEARS.” *Nature Biotechnology* 42 (6): 927–35.
- scVI-tools Development Team. 2024. “scVI User Guide: Differential Expression.” https://docs.scvi-tools.org/en/stable/user_guide/background/differential_expression.html.
- Theodoris, Christina V, Ling Xiao, Anant Chopra, Mark D Chaffin, Zeina R Al Sayed, Matthew C Hill, Helene Mantineo, et al. 2023. “Transfer Learning Enables Predictions in Network Biology.” *Nature* 618 (7965): 616–24.

Datasets

Dataset	Citation	Cell line	Number of perturbations
Norman	Norman et al. (2019)	K562	Single (148) + double (128)
Adamson	Adamson et al. (2016)	K562	Single (81)
Replogle	Replogle et al. (2022)	K562, RPE1	Single (1087, 1534)

Methods

For the benchmarks, we ran GEARS version 0.1.2 and scGPT version 0.2.1 with the recommended settings. For the double perturbation data by Norman et al. (2019), we assigned all single perturbations and half of the double perturbations to the training set and kept the other half of the double perturbations for validation. For the three single perturbations (Adamson et al. 2016; Replogle et al. 2022), we used GEARS’ *simulation* test-training assignment configuration. We ran all benchmarks twice initialized with different seeds. scGPT and our *linear* model can only predict perturbations which are in the genes and the *linear pretrained* model can only predict perturbations that are in the reference data, thus we only show perturbations in Figure 1 A where all methods made a prediction.

The additive predictions for the Norman data were calculated by first calculating the difference between each single perturbation and the control (i.e., unperturbed) condition. The prediction for a double perturbation then was the sum of the expression in the control condition plus the expression changes observed in both single perturbations. This meant that we did not use the double perturbations in the training set to improve the predictions.

We fitted the LEMUR model with two latent dimensions as we did not expect any relevant latent variation. We coded the single and double perturbations in the training set as a model matrix with 0/1 entries where each column was a perturbation target and a 1 indicated that a target had been perturbed in the cell.

Linear model

We trained the linear model for the single perturbation prediction on the responses to the set of training perturbations, which were CRISPR knockouts of genes. The training data are thus a matrix \mathbf{Y} of size $G \times C^{\text{tr}}$, where G is the length of the gene expression profiles and C^{tr} is the number of perturbations. We assume that \mathbf{Y} contains the variance stabilized

expression values aggregated across cells with the same condition, where the mean expression per gene in the control condition has been subtracted. We decompose

$$\mathbf{Y} = \mathbf{U} \text{diag}(d) \mathbf{V}^T$$

using the reduced-rank SVD decomposition, where \mathbf{U} is an orthonormal matrix of size $G \times P$, \mathbf{V} is an orthonormal matrix of size $C^{\text{tr}} \times P$, and d are the P singular values. The gene embedding $\mathbf{Z} = \text{diag}(d) \mathbf{U}^T$ is a low-dimensional representation which genes (i.e., rows of \mathbf{Y}) have similar expression changes under the training perturbations. From \mathbf{Z} we select the genes that also occur in the perturbation training set and fit a ridge regression to relate the gene embedding \mathbf{Z} back to the original expression space \mathbf{Y}

$$\arg \min_{\beta_0, \mathbf{B}} \left\| \sum_c^{C^{\text{tr}}} \mathbf{Y}_{:c} - \left(\beta_0 + \sum_{p=1}^P \mathbf{B}_{:p} \mathbf{Z}_{p,g=c} \right) \right\|^2 + \lambda \|\mathbf{B}\|^2,$$

where β_0 is a vector of length G and \mathbf{B} is a matrix of size $G \times P$.

As this model is just a simple linear regression, we can predict the gene expression changes for any perturbation c of a gene that is in the rows of \mathbf{Y}

$$\hat{\mathbf{Y}}_{:c} = \beta_0 + \sum_{p=1}^P \mathbf{B}_{:p} \mathbf{Z}_{p,g=c}.$$

For the transfer learning, we assume that we have a reference dataset \mathbf{Y}^{ref} of size $G \times C^{\text{ref}}$ where the perturbations in the columns of \mathbf{Y}^{ref} overlap with the training perturbations in \mathbf{Y} . We decompose

$$\mathbf{Y}^{\text{ref}} = \mathbf{T} \text{diag}(a) \mathbf{W}^T$$

using a rank-reduced SVD decomposition where \mathbf{T} is an orthonormal matrix of size $G \times P$, \mathbf{W} is an orthonormal matrix of size $C^{\text{ref}} \times P$, and a are the P singular values. The learned perturbation embedding $\mathbf{E}^{\text{ref}} = \text{diag}(a) \mathbf{W}^T$ of size $P \times C^{\text{ref}}$ encodes how similar the perturbations are in the reference data. We align the reference embedding to the embedding of the training perturbations $\mathbf{E}^{\text{tr}} = \text{diag}(d) \mathbf{V}^T$ using ridge regression

$$\arg \min_{\beta_0, \mathbf{B}} \left\| \sum_c^{C^{\text{tr}}} \mathbf{E}_{:c}^{\text{tr}} - \left(\beta_0 + \sum_{p=1}^P \mathbf{B}_{:p} \mathbf{E}_{p,c'=c}^{\text{ref}} \right) \right\|^2 + \lambda \|\mathbf{B}\|^2,$$

where β_0 is a vector of length G and \mathbf{B} is a matrix of size $G \times P$.

With this model, we can predict the gene expression changes for any perturbation c that is in the reference data

$$\hat{\mathbf{Y}}_{:c} = \mathbf{U} \left(\beta_0 + \sum_{p=1}^P \mathbf{B}_{:p} \mathbf{E}_{p,c'=c}^{\text{ref}} \right).$$

For the Replogle datasets in Figure 1 A, we pretrained the model on K562 to predict the RPE1 data and pretrained on RPE1 to predict the K562. The pretraining for the Adamson data was done on the K562 data. For both the *linear* and *linear pretrained* methods, we used $P = 5$ PCA dimensions and a ridge penalty of $\lambda = 0.1$.

Extended data figures

Figure 2: (A) Line plot showing the effect of varying PCA dimension (x axis) and ridge penalty (color) for the prediction performance on the Adamson and Replogle RPE1 data. The results are faceted performance in the test and training sets. The performance is measured by the average correlation between prediction and observed expression change compared to the control condition. (B) Same plot as (A), but measuring performance by distance between predicted and observed expression change. (C) Scatter plot of the the average correlation per perturbation (i.e., how good the predictions are) against the distance of the aligned K562 PCA position to the corresponding RPE1 PCA position (log-scale). The horizontal and vertical lines show the respective means for the correlation and distance.